# Mesolimbic dopamine release precedes actively sought aversive stimuli in mice

Yosuke Yawata[1], Yu Shikano[2], Jun Ogasawara[1], Kenichi Makino[1], Tetsuhiko Kashima[1], Keiko Ihara[2], Airi Yoshimoto [1], Shota Morikawa[1,3], Sho Yagishita[4], Kenji F. Tanaka [2] & Yuji Ikegaya [1,3,5] ✉

In some models, animals approach aversive stimuli more than those housed in an enriched environment. Here, we found that male mice in an impoverished and unstimulating (*i.e.*, boring) chamber without toys sought aversive air puffs more often than those in an enriched chamber. Using this animal model, we identified the insular cortex as a regulator of aversion-seeking behavior. Activation and inhibition of the insular cortex increased and decreased the frequencies of air-puff self-stimulation, respectively, and the firing patterns of insular neuron ensembles predicted the self-stimulation timing. Dopamine levels in the ventrolateral striatum decreased with passive air puffs but increased with actively sought puffs. Around 20% of mice developed intense self-stimulation despite being offered toys, which was prevented by administering opioid receptor antagonists. This study establishes a basis for comprehending the neural underpinnings of usually avoided stimulus-seeking behaviors.

When we have nothing of interest to attend to, we have an unpleasant emotion and often try to avoid that feeling[1,2]. This negative emotion[3–5] has not attracted enough attention in biological research, although it has important implications for emotional functions. In humans, this negative emotion is associated with mind wandering[3,6]. Moreover, the predisposition to a negative emotion[7] is correlated with the tendency for drug abuse[8,9], gambling disorder[10], and anxiety disorder[9,11]. Therefore, the predisposition to a negative emotion has a clinical aspect in the treatment of psychiatric disorders. However, the neural mechanism underlying the negative emotion that arises when there is nothing to do remains unknown, partly because it has mainly been investigated in humans, where invasive approaches, such as recording from single neurons or manipulating neuronal activity, are not largely applicable.

Recent studies have demonstrated that minks (*Neovison vison*) housed in normal cages are more likely to approach various stimuli than those housed in environmentally enriched cages[12–15]. These researchers speculate that this behavior reflects a boredom-like negative emotion. In the present study, we report that mice placed in an impoverished chamber actively approach aversive stimuli in the same way as humans who feel bored[16–18]. Using this animal model, we identified the insular cortex as a brain region related to actively seeking aversive stimuli. When actively sought, aversive stimuli, which should elicit a reduced release of rewarding dopamine, increased dopamine release. As a result, mice began to repeat the self-stimulation and eventually performed nose pokes intensely to experience the increase in dopamine. We also found that opioid receptor antagonists prevent the empty room-triggered transition to an intense behavioral state.

## Results

### Mice actively seek aversive stimuli in an impoverished chamber

We examined whether mice exhibit aversive stimulus-seeking behaviors as reported in previous studies in which humans deliver aversive electric shocks to themselves in monotonous situations[16–18]. A mouse was placed in a 30×20 cm chamber that had a nose-poke hole in its wall (Fig. 1a). When a mouse poked its nose into the hole, it was

---

[1]Graduate School of Pharmaceutical Sciences, The University of Tokyo, Tokyo 113-0033, Japan. [2]Division of Brain Science, Keio University School of Medicine, Tokyo 160-8582, Japan. [3]Institute for AI and Beyond, The University of Tokyo, Tokyo 113-0033, Japan. [4]Center for Disease Biology and Integrative Medicine, Faculty of Medicine, The University of Tokyo, Tokyo 113-0033, Japan. [5]Center for Information and Neural Networks, National Institute of Information and Communications Technology, Osaka 565-0871, Japan. ✉e-mail: yuji@ikegaya.jp

 

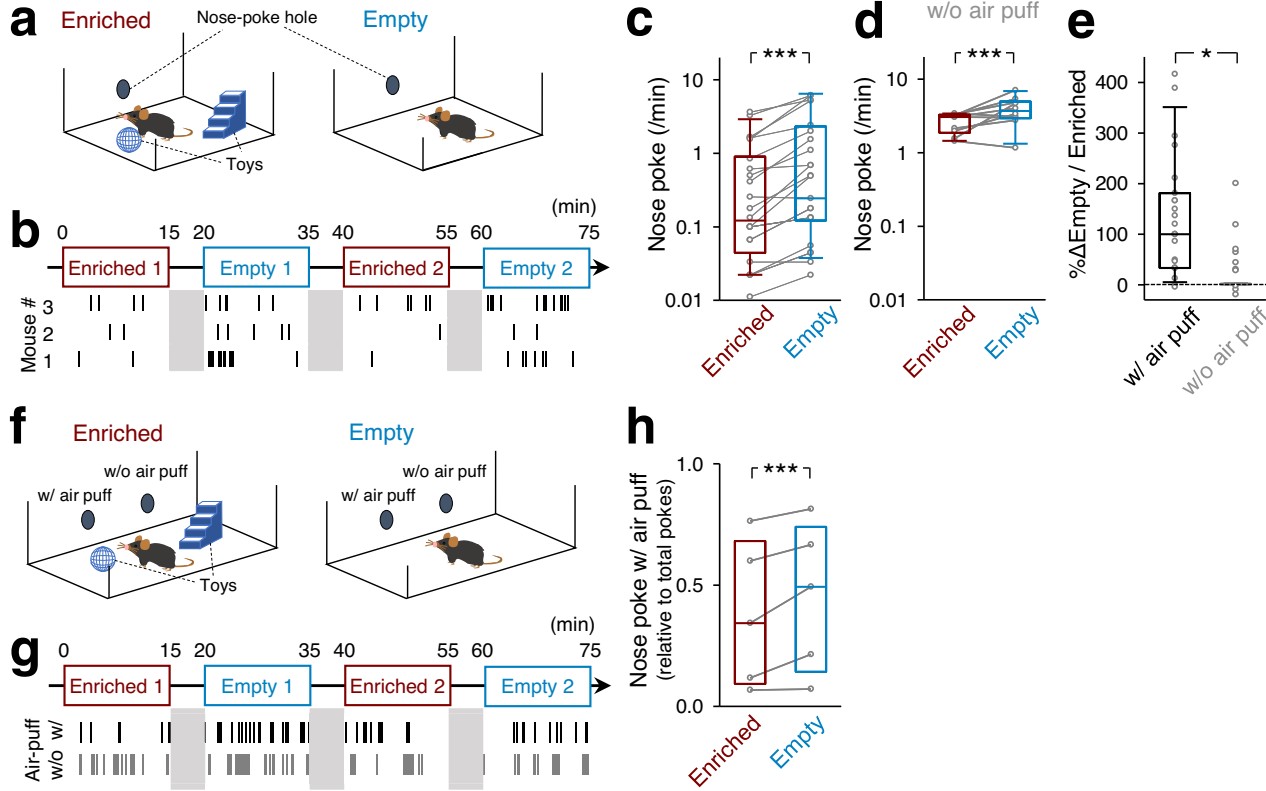

**Fig. 1 | Mice received air-puff stimuli more frequently in an empty chamber.**
**a** The experimental design for our behavioral test. When mice poked their noses into a hole on a chamber wall, they received a brief air-puff stimulus. The chamber was equipped with (enriched) or without (empty) toys. **b** Time schedules of the behavioral test. Each mouse was alternately placed in an enriched or empty chamber for 15 min each with an interval of 5 min. The bottom raster plot indicates the timings of nose pokes of three representative mice. **c** Frequencies of nose pokes in enriched and empty rooms. The box plots indicate the median, inter-quartile range, and extreme values. Each line indicates a single mouse, and the values at both ends represent the averages of two sessions. $^{***}P < 0.0001$, $n = 23$ mice, two-sided paired bootstrap test. **d** The same as (c) but for the without-air puff conditions. $^{***}P < 0.0001$, $n = 10$ mice, two-sided paired bootstrap test. **e** Percent

increases in the ratios of the numbers of nose pokes in the empty sessions to those in the enriched sessions are compared between chambers with and without air puffs. The box plots indicate the median, interquartile range, and extreme values. $^{*}P = 0.043$, $n_{w/air\ puff} = 23$ mice, $n_{w/o\ air\ puff} = 10$ mice, two-sided two-sample boot-strap test. **f** An experimental design for a two-hole behavioral test. One hole in a chamber wall resulted in air-puff stimulation upon nose-poking, whereas the other did not. **g** Time schedules of the behavioral test and the timings of nose pokes of a representative mouse. **h** The ratios of the numbers of nose pokes into the hole with air puffs to the total number of nose pokes into both holes. The box plots indicate the median and interquartile range. $^{***}P < 0.0001$, $n = 5$ mice, two-sided paired bootstrap test.

immediately subjected to an air-puff stimulus to the face. The air puffs were considered inherently aversive[19–21]. Indeed, we confirmed that in a passive avoidance test, mice avoided air puffs by staying in a lighted room after they learned that they would experience air puffs if they entered a dark room (Supplementary Fig. 1, $P < 0.01$, $n = 6$ mice each, two-sample bootstrap test).

We investigated whether mice actively seek air puffs when they are placed in an impoverished environment. We prepared two experimental conditions (Fig. 1a): an 'enriched' chamber equipped with a ladder and seesaw and an impoverished (hereinafter referred to as 'empty') chamber with no toys. Mice experienced the enriched and empty sessions alternately, twice for 15 min each, and we counted the numbers of nose pokes in each session (Fig. 1b). We found that the frequencies of nose pokes were, on average, 2.1 times higher in the empty sessions than in the enriched sessions (Fig. 1c, $P < 0.0001$, $n = 23$ mice, paired bootstrap test, Supplementary Movie 1), suggesting that mice were more likely to seek air puffs in the empty chamber. This trend was not altered depending on which chamber (i.e., empty or enriched) mice experienced first (Supplementary Fig. 2a–d). The frequencies of nose pokes increased gradually during an empty session lasting 15 min (Supplementary Fig. 2e, $P = 0.011$, $0.0024$, and $0.014$ for 0–3 min versus 3–6, 6–9, and 9–12 min, respectively, $n = 23$ mice, paired bootstrap test with Bonferroni correction), suggesting that the

motivation to seek air puffs accumulated during an empty session. However, the frequencies of nose pokes did not increase within a day or throughout the 3-d tests (Supplementary Fig. 2d, $P = 0.248$ and $0.313$ for enriched and empty conditions, respectively, $n = 23$ mice, Jonckheere trend test). Therefore, the effect of novelty seeking (or the learning process) on nose poking was likely minimal. This is probably because mice were habituated to the chamber prior to the behavioral test.

To rule out the possibility that mice spent less time nose-poking in the enriched chamber simply because they played with the toys, we repeated the same test using a 'w/o air-puff' chamber in which nose-poking did not trigger air-puff stimulation. In the w/o air-puff conditions, the overall frequency of nose pokes was higher than that in the chambers in which mice received air puffs (cf., Fig. 1c, d), and mice exhibited more nose pokes in the empty sessions than in the enriched sessions (Fig. 1d, $P < 0.0001$, $n = 10$ mice, paired bootstrap test). However, the ratios of the numbers of nose pokes in the empty sessions to those in the enriched sessions were still higher in the air-puff conditions than in the w/o air-puff conditions (Fig. 1e, $P = 0.043$, $n_{w/air\ puff} = 23$ mice, $n_{w/o\ air\ puff} = 10$ mice, two-sample bootstrap test). Thus, even considering the time lost due to playing with toys, mice still performed nose pokes more actively in the empty sessions. The distance traveled in the w/air-puff conditions was longer than that in the

w/o air-puff conditions (Supplementary Fig. 3, $P = 8.0 \times 10^{-4}$ and 0.0012 for enriched and empty conditions, respectively, two-sampled bootstrap test with Bonferroni correction). These results rule out the possibility that aversive air puffs decreased the exploratory behavior of mice. In addition, we prepared a two-hole chamber in which only one hole was associated with air-puff stimulation while the other was blank (Fig. 1f). The ratios of the number of noses poke into the hole in which mice received air-puffs to the total number of nose pokes were higher in the empty sessions than in the enriched sessions (Fig. 1g, h, $P < 0.0001$, $n = 5$ mice, paired bootstrap test). These results indicate that the decrease in nose poking in the enriched session cannot be explained solely by the fact that the mice played with toys. Therefore, we concluded that mice preferred air puffs under empty conditions.

The negative emotion that arises when there is nothing to do is characterized as a high-arousal state, which may be associated with an activated sympathetic nervous system, characterized by occurrences such as higher heart rates[22] and higher respiratory rates[23–25]. During behavioral tests, we recorded electrocardiograms; local field potentials (LFPs) from the olfactory bulb, in which the respiratory rhythms were reflected[23,25]; and electromyograms from the dorsal neck muscles (Supplementary Fig. 4). We plotted the time change in heart rates, their variabilities (HRVs, coefficients of variation in RR intervals), and respiratory rates relative to the timings of nose pokes (Supplementary Fig. 4). HRVs serve as a biomarker of parasympathetic activity and decrease under stressful conditions[26,27]; in fact, the HRVs in the empty conditions were higher in the home cage (Supplementary Fig. 5, $P = 0.0257$, $n_{Empty} = 43$, $n_{home\ cage} = 18$, one-sided Kolmogorov–Smirnov test), suggesting that an impoverished environment is stressful for mice. The mean heart rates before nose pokes were significantly reduced after nose pokes (Supplementary Fig. 6a, b, $P = 0.017$, $n = 294$ events from 6 mice, paired bootstrap test), whereas the mean HRVs were increased after nose pokes (Supplementary Fig. 6a, c, P = 0.013). The mean respiratory rates were reduced after nose pokes (Supplementary Fig. 2b, c, $P = 0.0052$, $n = 263$ events from 4 mice). The electromyogram did not exhibit a comparable long-lasting change in response to nose pokes (Supplementary Fig. 4d, e, $P = 0.39$, $n = 294$ events from 6 mice). Thus, the stressful states were likely alleviated by air puffs. Taken together, our data are consistent with the hypothesis that mice, like humans, have negative emotions in an empty chamber and try to avoid this aversive situation.

## The insular cortex is involved in aversive stimulus-seeking behavior

To explore the brain region that is related to aversive stimulus-seeking behavior, we focused on the agranular insular cortex (AIC) because a human imaging study demonstrated that the AIC is activated in a boring situation[28]. When AIC neurons were deactivated by bilateral injection of a cocktail of gamma-aminobutyric acid (GABA) receptor agonists, muscimol and baclofen (Mus+Bac) (Fig. 2a and Supplementary Fig. 7a), the frequencies of nose pokes in the empty sessions were decreased to levels not different from those in the enriched sessions (Fig. 2b, $P < 0.0001$, $n = 12$ mice, paired bootstrap test with Bonferroni correction). However, Mus+Bac injection also reduced locomotor activity in the test chamber (Supplementary Fig. 8a, $P < 0.0001$, $n = 12$ mice). Thus, it was possible that the lowered locomotion or behavioral motivation caused the reduction in nose-poking behavior. To address this possibility, we divided the nose-poke frequency by the distance traveled in the behavioral test and found that the poking frequency per unit of distance was still reduced by Mus+Bac injection (Supplementary Fig. 8b, $P = 0.008$, $n = 12$ mice). Similar results were obtained when the nose-poke frequency was divided by the total active time, defined as periods when the movement speed exceeded 10 cm/s (Supplementary Fig. 8c, d, $P = 0.033$, $n = 12$ mice, paired bootstrap test with Bonferroni correction). Because the effects of Mus+Bac injection were still confirmed after these corrections, we concluded that nose pokes

reduced by AIC deactivation cannot be fully explained by the reduced locomotion alone. Moreover, although Mus+Bac injection reduced locomotion in the test chamber, the injection did not affect locomotion in an unfamiliar wide-open field (Supplementary Fig. 8e, $P = 0.43$, $n_{Saline} = 7$ mice, $n_{Mus+Bac} = 10$ mice, two-sample bootstrap test). Thus, it is unlikely that AIC inactivation reduced motivation, thereby resulting in a decrease in nose pokes. In control experiments, Mus+Bac was injected into the posterior parietal cortex (Supplementary Fig. 7b), but an injection in this region did not affect the nose-poke frequency (Supplementary Fig. 8f, $P = 0.27$, $n = 5$ mice, paired bootstrap test with Bonferroni correction).

## Gamma oscillations in the AIC increase before nose pokes

To address the neural representation of aversive stimulus-seeking behavior, we recorded LFPs from the AIC (Fig. 2c and Supplementary Fig. 7c). Gamma-band oscillations (30–150 Hz) increased over 30 s before nose pokes (Fig. 2d, e, $P = 0.0053$, $n = 263$ events, Jonckheere trend test) and returned to the baseline level after nose pokes. The prepoke increase in gamma oscillations may represent aversive states in mice, as previous studies have demonstrated that gamma oscillations in the insular cortex are induced by nociceptive stimuli or anticipation of pain[29,30]. This idea led to the possibility that increasing AIC gamma oscillations may cause an increase in nose pokes. In fact, the AIC gamma oscillations increased steeply in the last quarter of the nose-poke events (Supplementary Fig. 9a, $P = 0.0034$, $n = 18$ events, Jonckheere trend test). The gamma oscillations in the enriched sessions were higher than those in the empty sessions ($P = 0.011$, $n_{Enriched} = 39$ events, $n_{Empty} = 134$ events, two-sample Kolmogorov–Smirnov test) and increased in association with nose pokes significantly in the enriched sessions and marginally in the empty sessions (Supplementary Fig. 9b, $P = 0.0079$, $n_{Enriched} = 39$ events, $P = 0.051$, $n_{Empty} = 134$ events, Jonckheere trend test). This gamma increase might reflect the fact that stronger motivation to seek aversive stimuli was required in the enriched sessions.

To examine whether increasing AIC gamma oscillations increase nose pokes, we optogenetically induced gamma oscillations using mice in which channelrhodopsin 2 (ChR2) was expressed in AIC neurons (Supplementary Fig. 7d). When the AIC in neocortical slice preparations was illuminated by blue-light pulses at 40 Hz (Fig. 2f and Supplementary Fig. 10), the power of gamma-band oscillations was increased in the AIC LFP, and depolarization and intermittent firing that were locked at 40 Hz were observed in whole-cell current-clamp recorded neurons (Supplementary Fig. 10a–c). Under these optogenetic stimulation conditions, mice stimulated at a frequency of 40 Hz in the empty sessions exhibited nose pokes more frequently than nonstimulated mice (Fig. 2g, $P = 0.020$, $n = 7$ mice, paired bootstrap test). However, optogenetic stimulation was not a direct motor trigger for nose poking because nose pokes were not time-locked to the 40-Hz illumination (Supplementary Fig. 10d, e). Rather, AIC gamma oscillations were likely to increase the motivation to seek air puffs. In the enriched session, 40-Hz stimulation did not increase nose pokes (Supplementary Fig. 10f, $P = 0.44$, $n = 7$ mice, paired bootstrap test). Although stimulated mice exhibited slightly lower locomotor activity (Supplementary Fig. 10g, $P = 0.018$, $n = 7$ mice, paired bootstrap test), their nose-poke frequencies per unit of distance traveled were still significantly higher than those of the nonstimulated mice (Supplementary Fig. 10h, $P = 0.021$, $n = 7$ mice, paired bootstrap test). Similar results were obtained when the nose-poke frequencies were divided by the active time periods (Supplementary Fig. 10i, j, $P = 0.037$, $n = 7$ mice, paired bootstrap test). The distance traveled in the open field test was not affected by 40-Hz stimulation (Supplementary Fig. 10k, $P = 0.86$, $n_{No\ stim.} = 5$ mice, $n_{Stim} = 7$ mice, two-sample bootstrap test). When the AIC was illuminated at 10 Hz, nose-poke frequencies were not changed (Supplementary Fig. 10l, $P = 0.11$, $n = 7$ mice, paired bootstrap test). Thus,

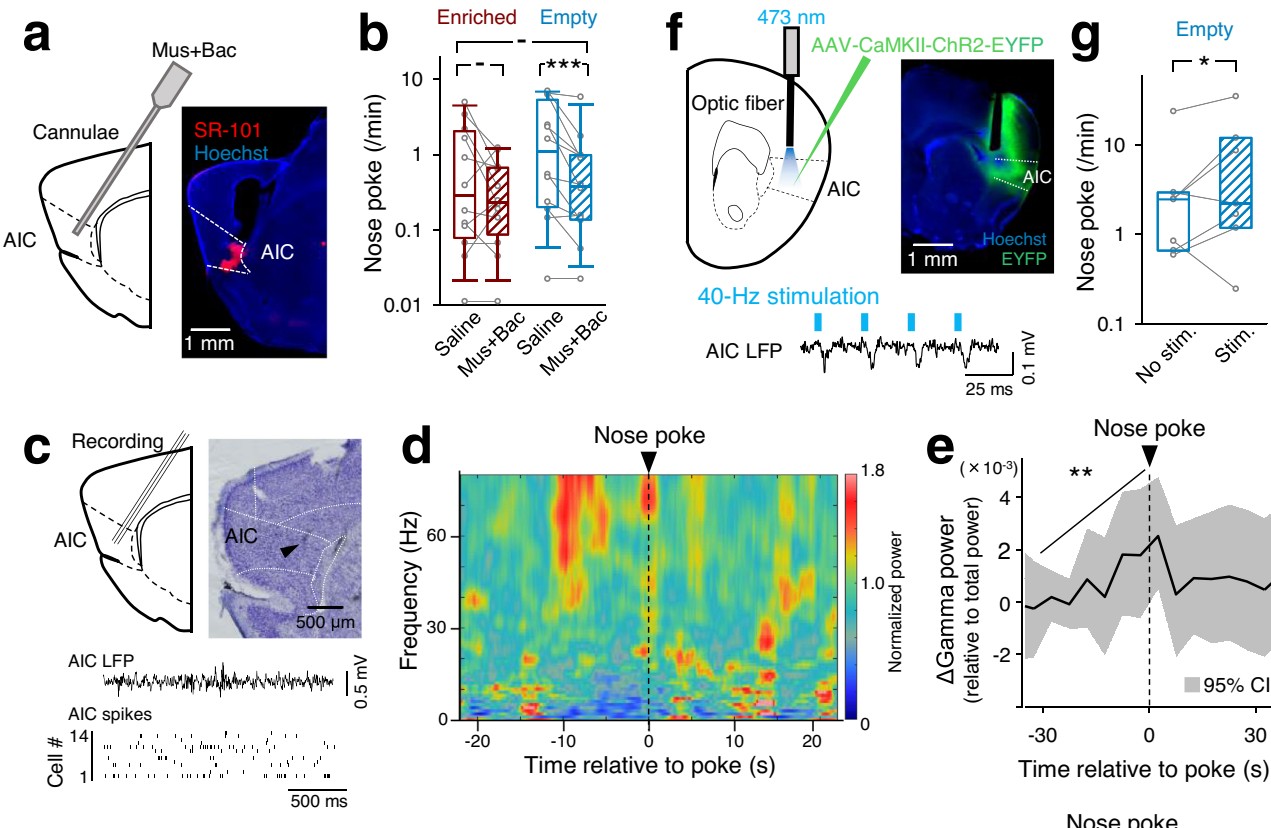

**Fig. 2 | Gamma oscillations in the AIC increase before nose-poking behavior.**
**a** Muscimol and baclofen (Mus+Bac) were injected into the AIC together with the fluorescent dye SR-101, by which the tip of cannulas for drug injection was confirmed *post hoc*. Similar macrographs were obtained from independent experiments performed on 12 mice. **b** The frequencies of nose pokes in enriched and empty chambers were compared between mice treated with saline or Mus+Bac. The box plots indicate the median, interquartile range, and extreme values. `P > 0.17, ***P < 0.0001, n = 12 mice, two-sided paired bootstrap test with Bonferroni correction. **c** Extracellular recording from the AIC. The arrowhead indicates the location of the electrode tip. The bottom panel demonstrates a representative trace of local field potentials (LFPs) and a raster plot of spikes of 14 simultaneously recorded neurons. Similar macrographs were obtained from independent experiments

performed on 4 mice. **d** Representative power spectrogram of AIC LFPs before and after a nose poke. The LFP power was normalized for each frequency across the total recording period and plotted on a pseudocolor scale. **e** Time change in the gamma oscillation (30–150 Hz) power around nose pokes. The thick line and gray areas represent the means and the 95% confidence intervals (CIs) defined by a one-sample bootstrap test for 263 events. **P = 0.0053, one-sided Jonckheere trend test for the −30-to-0-s period. **f** Viral injection of AAV-CaMKIIa-ChR2-EYFP into the AIC (top). Similar macrographs were obtained from independent experiments performed on 7 mice. Blue-light pulses at 40 Hz for 60 s were applied every 120 s. **g** The frequencies of nose pokes in an empty room were increased in mice with blue-light stimulation. The box plots indicate the median and interquartile range. `P = 0.020, n = 7 mice` two-sided paired bootstrap test.

artificially induced AIC gamma oscillations can modulate nose-poking behaviors.

## Firing patterns of AIC neurons predict nose pokes

Given that the AIC is involved in aversive stimulus-seeking behaviors, we assumed that nose-poking is decodable from the unit activity of AIC neurons. By sorting spikes based on multiple features of the spike waveform (see Methods, Supplementary Fig. 11a, b), we identified a total of 2,689,887 spikes of 93 putative excitatory AIC neurons during the behavioral test. We visualized the firing patterns (time-series vectors of the firing rates of individual neurons) of simultaneously recorded AIC neurons around nose-poke timings (Supplementary Fig. 12a) using principal component analysis, a dimension reduction technique (Supplementary Fig. 12b). The time evolution of the firing patterns in the first three principal components indicated a characteristic orbit around nose-poke timing, suggesting that the AIC firing patterns were correlated at least in part with nose-poke behavior. Furthermore, using a linear support vector machine, we sought to classify whether the firing pattern at a given time point (2-s bin) was associated with a nose poke. The classification accuracy exceeded the stochastic level of 50% for a period of approximately 5 s before and after nose pokes (Supplementary Fig. 12c, n = 8, one-sample bootstrap

test), suggesting that AIC neuron ensembles encoded the timings of nose pokes. We then examined the nose poke-relevant responses of AIC neurons at the single-cell level. Out of a total of 93 neurons recorded, 42 neurons (45.2%) responded significantly to nose pokes; 10.8% and 12.9% increased their firing rates before and after nose pokes, respectively, whereas 11.8% and 9.7% decreased the firing rates before and after nose pokes, respectively (Supplementary Fig. 11c, d).

## Dopamine is released in the ventral striatum before nose pokes are performed

To estimate how mice perceive air puffs when they actively receive them, we monitored the extracellular level of dopamine in the ventrolateral striatum (VLS)[31–33] using in vivo fiber photometry to detect the fluorescent signal of GRAB_DA2m, a genetically encoded indicator for dopamine[34] (Fig. 3a and Supplementary Fig. 7e). In control experiments, we confirmed that food consumption increased the levels of dopamine in the VLS (Fig. 3b, P < 0.0001, n = 8 mice, one-sample bootstrap test), suggesting that foods served as rewards[32,35,36]. In contrast, when air was puffed into the faces of mice (irrelevant to nose pokes), the VLS dopamine levels briefly increased and then decreased (passive air puff in Fig. 3c–e, n = 8 mice). The late decrease in dopamine is thought to represent aversion to air puffs[37], whereas the early

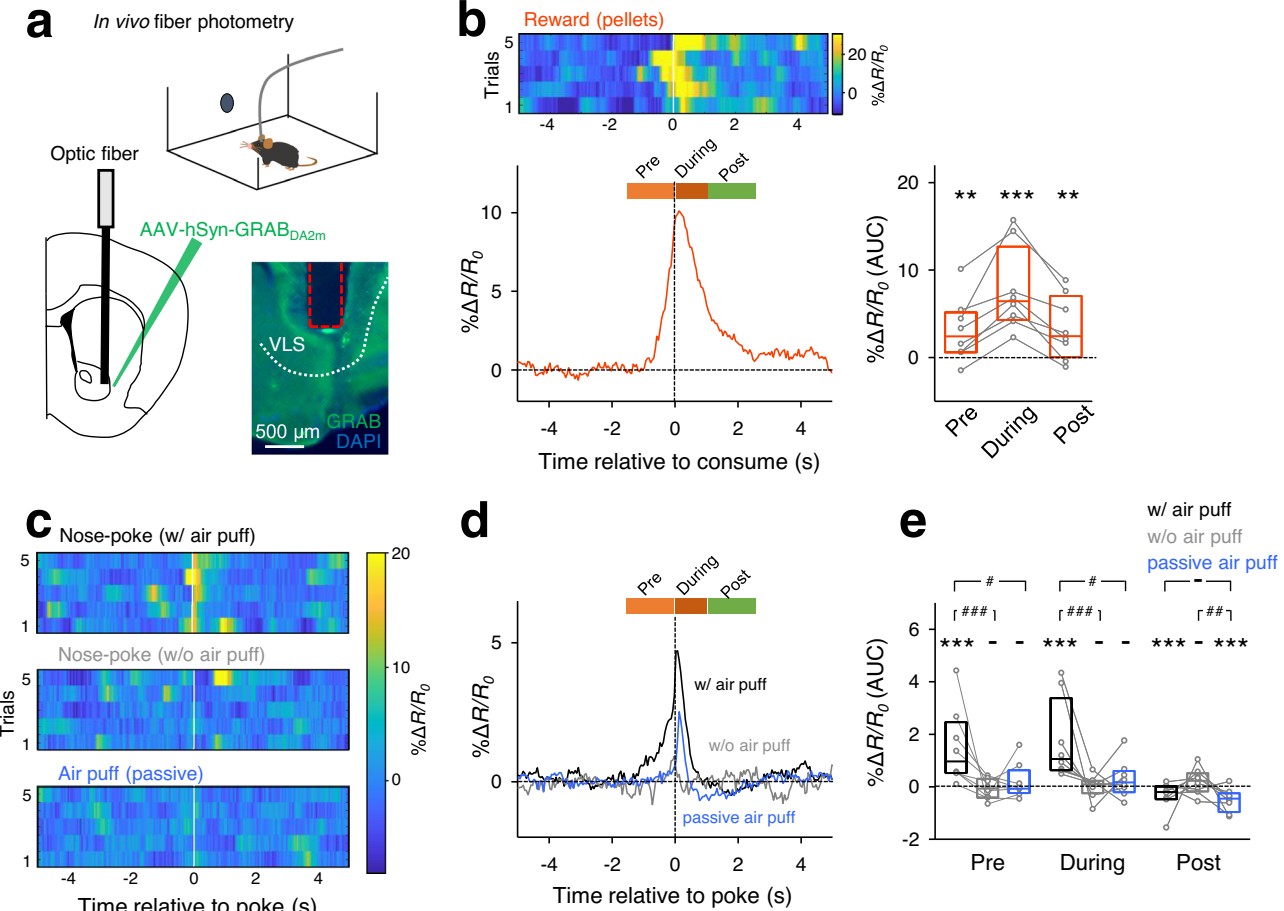

**Fig. 3 | Dopamine is released in the ventrolateral striatum before nose pokes.** **a** GRAB$_{DA2m}$, a fluorescent dopamine indicator, was expressed in neurons of the ventrolateral striatum and the fluorescent signals were optically recorded through an optic fiber during the behavioral test. **b** *Top*: Representative heatmap demonstrating that the GRAB$_{DA2m}$ signal increased during the consumption of food rewards. The pseudocolor scale represents the fluorescence intensity change ($\Delta R/R_0$) over the baseline. *Bottom left*: Mean $\Delta R/R_0$ aligned to the onsets of 13−29 pellet consumptions in 8 mice. The orange, red, and green bars above the plot indicate the preconsumption period, the during-consumption period, and the post-consumption periods, respectively. *Bottom right*: The area under the curves (AUCs) of $\Delta R/R_0$ during these three periods. The box plots indicate the median and interquartile range. **P = 0.0034, 0.001 for the prepoke and the postpoke period, respectively. ***P < 0.0001, n = 8 mice, two-sided one-sample bootstrap test. **c** Representative heatmaps for GRAB$_{DA2m}$ signals relative to the onsets of nose pokes with air puffs (top), nose pokes without air puffs (middle), and passive air puffs (bottom). **d** Mean $\Delta R/R_0$ aligned to the nose-poke onsets with and without air puffs (black and gray, respectively) and to passive air puffs (blue) of 8 mice. **e** AUC of $\Delta R/R_0$ during the prepoke period (left), the during-poke period (middle), and the postpoke period (right). The black, gray, and blue box plots indicate the data of nose pokes with air puffs, without air puffs and with passive air puffs, respectively. The box plots indicate the median and interquartile range. $\dot{}$P = 0.55, 0.42, 0.89, 0.22, 0.21 for the prepoke period in w/o air-puff condition, in passive air-puff condition, the during period in w/o air-puff condition, in passive air-puff condition, and the postpoke period in w/o air-puff condition, respectively. ***P < 0.0001, n = 8 mice, one-sample bootstrap test. #P = 0.038, 0.022, for the prepoke period in w/ air-puff condition versus passive air-puff condition, the during period in w/ air-puff condition versus passive air-puff condition, respectively. ##P = 0.0018 for the postpoke period in w/ air-puff condition versus passive air-puff condition, ###P < 0.0001, two-sided paired bootstrap test with Bonferroni correction.

increase in dopamine resulted from stimulus saliency[38]. When mice actively performed nose pokes to receive air puffs, dopamine levels increased before and during nose pokes, followed by a delayed decrease in dopamine (nose pokes in Fig. 3c−e, P < 0.0001, n = 8 mice, one-sample bootstrap test). The increases before and during nose pokes were greater than those induced by passive air puffs (Fig. 3c−e, P = 0.025, n = 8 mice), whereas the late decrease did not differ between active and passive air puffs (P > 0.99, n = 8 mice, paired bootstrap test with Bonferroni correction). To examine whether dopamine levels reflect nose-poke behavior (*i.e.*, animal's movement itself[39]), we monitored dopamine levels when mice poked their noses into a blank hole that did not deliver air puffs. In that w/o air-puff condition, dopamine levels were not changed (Fig. 3c−e, P = 0.46, 0.81, and 0.35, for pre-, during-, and postpoke periods, respectively, n = 8 mice, one-sample bootstrap test). In addition, neither the velocity in the baseline period (4 s to 2 s before the nose pokes) nor that in the prepoke period was

significantly correlated with the dopamine level (Supplementary Fig. 13a, b). These suggest that dopamine increase does not reflect a mere movement such as a nose poke and other preparation behavior. Therefore, the anticipation of an air puff may act as a reward, while air puffs themselves remain aversive.

## Aversive stimulus-seeking behavior sometimes triggers intense self-stimulation

Through this series of experiments, we found that the frequency of nose pokes varied greatly from mouse to mouse and that a portion of mice performed an extremely high number of pokes; note that the vertical axis in Fig. 1c is plotted on a logarithmic scale. In fact, these mice repeated air-puff stimulation many times in both the enriched and empty sessions (Fig. 4a and Supplementary Movie 2). The nose-poke frequency, as a whole, showed a local minimum at 2.4 pokes/min (Supplementary Fig. 14a); therefore, nose-poking at frequencies of

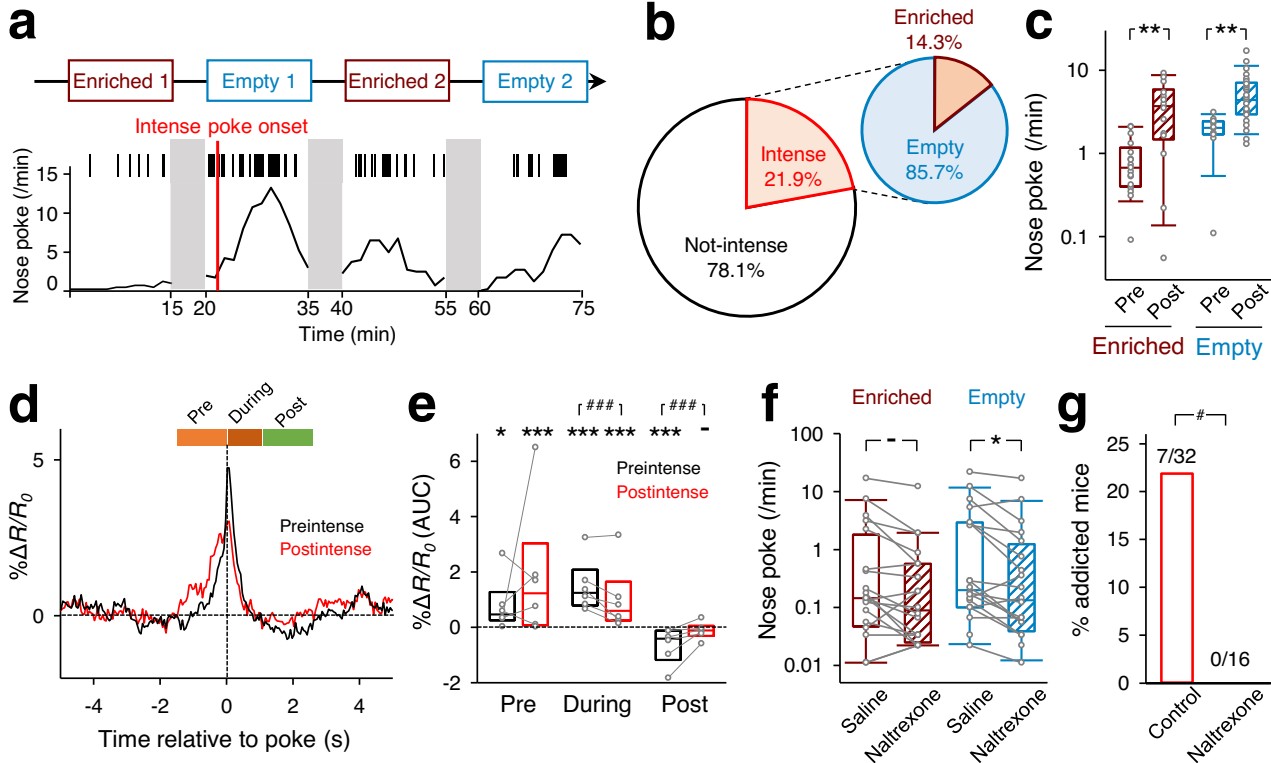

**Fig. 4 | Pharmacological blockade of opioid receptors prevents intense air-puff seeking to nose pokes. a** Data from a representative mouse that exhibited intense nose-poking behavior. The red line indicates the onset of intense behavior (threshold: 2.4 pokes/min). **b** The left graph indicates the percentage of mice that exhibited intense behavioral states, and the right graph indicates the percentage of the session types in which mice exhibited intense behavioral states for the first time (right). $n = 32$ mice. **c** Frequencies of nose pokes in enriched (dark red) and empty (blue) chambers before (pre) and after (post) the onset of an intense behavioral state. The box plots indicate the median, interquartile range, and extreme values. $^{**}P = 0.0020, 0.0044$ for the enriched and empty condition, respectively., $n_{\text{Enriched\_preintense}} = 21$ sessions, $n_{\text{Enriched\_postintense}} = 25$ sessions, $n_{\text{Empty\_preintense}} = 11$ sessions, $n_{\text{Empty\_postintense}} = 40$ sessions, two-sided two-sample

bootstrap test. **d** Mean $\Delta R/R_0$ of GRAB$_{DA2m}$ signals under the preintense (black) and postintense (red) conditions. **e** AUC of $\Delta R/R_0$ during the prepoke period (left), the during-poke period (middle), and the postpoke period (right). The black and orange box plots indicate the preintense and postintense conditions, respectively. The box plots indicate the median and interquartile range. $^{*}P = 0.026$, $^{*}P = 0.015$, $^{***}P < 0.0001$, $n = 6$ mice, two-sided one-sample bootstrap test. $^{*}P = 0.26$, $^{##}P < 0.01$ $n = 6$ mice, paired bootstrap test. **f** Frequencies of nose pokes in mice treated intraperitoneally with saline or 10 mg/kg naltrexone. The box plots indicate the median, interquartile range, and extreme values. $^{*}P > 0.99$, $^{*}P = 0.034$, $n = 12$ mice, two-sided paired bootstrap test with Bonferroni correction. **g** Fraction of mice that exhibited intense behavioral states. $n_{\text{Control}} = 32$ mice, $n_{\text{Naltrexone}} = 16$ mice, $^{#}P = 0.043$, Chi-square test.

more than 2.4 pokes/min was defined herein as an 'intense' state[40–43]. According to this definition, 7 of 32 mice (21.9%) experienced intense states. For 6 of the 7 mice (85.7%), the first intense states occurred in the empty sessions (Fig. 4b). Once the intense state occurred, high nose-poke frequencies were maintained in the following sessions, as evidenced by the number of nose pokes in a given empty session being positively correlated with a change in the number of nose pokes between two consecutive enriched sessions that were separated by the empty session (Supplementary Fig. 14b, R = 0.74, $P = 1.04 \times 10^{-13}$, $n = 55$ sessions, Pearson correlation coefficient). Moreover, mice that had experienced intense states at least once thereafter exhibited more pokes both in empty and enriched sessions than mice in 'preintense' states (Fig. 4c, P < 0.01, $n_{\text{Enriched\_preintense}} = 21$ trials, $n_{\text{Enriched\_postintense}} = 25$ trials, $n_{\text{Empty\_preintense}} = 11$ trials, $n_{\text{Empty\_postintense}} = 40$ trials, two-sample bootstrap test). In 'postintense' states, the VLS dopamine increase during the during-poke period was smaller than at the preintense period (Fig. 4d, e, P = 0.0022, $n = 6$ mice, paired bootstrap test), and the dopamine decrease during the post-poke period was no longer observed (Fig. 4d, e, P = 0.26, $n = 6$ mice, one-sample bootstrap test).

Finally, we administered naltrexone, an opioid receptor antagonist that is used for the treatment of alcohol use disorder[44,45], to mice intraperitoneally before the behavioral test. Naltrexone decreased the frequency of nose pokes in the empty sessions compared to that

observed in response to saline (Fig. 4f, P = 0.034, $n = 12$ mice, paired bootstrap test). The distances traveled by naltrexone-treated mice and the active times spent performing locomotion were also decreased (Supplementary Fig. 14a, P = 0.0064, $n = 19$ mice, paired bootstrap test with Bonferroni correction), but the nose-poke frequencies per unit of distance traveled and the active time periods were still significantly decreased by naltrexone (Supplementary Fig. 15b–d, P = 0.0052, P = 0.0068). The distance traveled in the open field test was not affected by naltrexone (Supplementary Fig. 15e, P = 0.82, $n_{\text{Control}} = 9$ mice, $n_{\text{Naltrexone}} = 10$ mice, two-sample bootstrap test), suggesting that the reduction in nose pokes by naltrexone was not only due to a reduction in locomotor activity per se. Importantly, none of the naltrexone-treated mice developed intense states (Fig. 4g, $n_{\text{Control}} = 32$ mice, $n_{\text{Naltrexone}} = 16$ mice, P = 0.043, $\chi^2$ test). In fact, dopamine release in the VLS before nose pokes was decreased by naltrexone administration (Supplementary Fig. 16a, b, P = 0.25, $n_{\text{Naltrexone}} = 4$ mice, one-sample bootstrap test), suggesting that the absence of dopamine release before nose pokes prevents mice from developing intense states.

## Discussion
Inspired by the fact that humans actively tolerate aversive stimuli when they are in a monotonous environment[16–18], we discovered that mice also actively engaged in nose-poking behavior that elicited air puffs.

Some of the mice exhibited excessive nose pokes as if they preferred air puffs. This idea is consistent with our findings that air puffs that were actively sought induced an increase in VLS dopamine release. Indeed, adult humans enjoy bitter (i.e., inherently aversive) drinks, such as coffee and beer. Some people may even prefer to purposefully experience pain. These paradoxical preferences are shaped by *posteriori* learning because children naturally avoid these stimuli. Our work may provide insight into the neural mechanism underlying masochistic-like behavior.

Human imaging studies have investigated the neural correlates of boredom, suggesting that some regions, such as the insular cortex, the prefrontal cortex, the precuneus, and the posterior cingulate cortex, respond to boredom conditions[28,46–48]. Our data demonstrated that the insular cortex is involved in aversive stimulus-seeking behavior in mice. The insular cortex is involved in interoception[49–53], the sense of the internal state of the body, such as visceral and cardiovascular signals[54–56]. Rodent studies suggest that the activity of insular neurons is associated with hunger and thirst[57–59] and is correlated with heart rates after fear conditioning[60]. Of note, we found that gamma oscillations in the AIC gradually increased before nose pokes were performed and that heart rates and respiratory rates decreased after nose pokes were performed. Therefore, we speculate that the internal state of seeking aversive stimuli is expressed in systemic signals, such as heartbeat and respiration, which are transmitted to the insular cortex as interoceptive information. The insular cortex is also known as the center of the salience network[51,61,62], which represents important information in the environment and mediates switching between the default mode network and the central executive network[63]. In fact, insular cortical activity is anti-correlated with the default mode network when boredom is induced in humans[46]. Therefore, it is possible that the nose-poke-related activity of AIC neurons in mice is associated with an attempt to engage in nose-poking to relieve stressful states.

One possible mechanism that drove mice into intense states is the activation of the reward system via insular cortical activation. Pertinently, the insular cortex is related to drug addiction[64–68]. Optogenetic activation of the insular cortex increases reward-related place preference indirectly by recruiting dopaminergic neurons in the ventral tegmental area[69]. Moreover, optogenetic stimulation of a subclass of AIC neurons increases dopamine release in the ventral striatum and promotes motivational vigor even in trials without reward[57]. Such dopamine elevation may contribute to the formation of what is referred to as a "prioritized salience map" in the parietal cortex[70,71]. In other words, dopamine modification may induce a preference for high-salience locations. Another line of evidence indicates that a transient drop in the dopamine level plays a role in associative learning of reward-related signals and that activating the ventral tegmental area eliminates the dopamine drop, resulting in a failure to discriminate reward cues[72]. Therefore, it is plausible that boredom-induced activation of the insular cortex eliminates an air-puff-induced decrease in dopamine, resulting in intense nose-poking behavior. Although we did not specify the brain region where naltrexone exerted its effect, it is possible that the effect was mediated by blockade of opioid receptors in the AIC[73] or reward-responsive regions, including the nucleus accumbens and the ventral pallidum[74]. In addition, our data showed that approximately half of AIC neurons responded to nose pokes (Supplementary Fig. 11). Another study suggested that a specific set of AIC neurons regulate motivational vigor and striatal dopamine via the nucleus tractus solitarii[57]. Thus, cell type-specific recording and manipulation of neural activity from multiple brain regions during boredom-relevant behaviors may reveal the neural mechanisms of boredom in more detail and thereby the pathogenic aspects of reward-related psychiatric disorders, including addiction[8–10], self-injury[16,17], and masochistic personality disorder.

## Methods

### Animals

Animal experiments were performed with the approval of the Animal Experiment Ethics Committee at The University of Tokyo (approval number: P29-11) and according to the University of Tokyo guidelines for the care and use of laboratory animals. These experimental protocols were carried out in accordance with the Fundamental Guidelines for Proper Conduct of Animal Experiment and Related Activities in Academic Research Institutions (Ministry of Education, Culture, Sports, Science and Technology, Notice No. 71 of 2006), the Standards for Breeding and Housing and Pain Alleviation for Experimental Animals (Ministry of the Environment, Notice No. 88 of 2006) and the Guidelines on the Method of Animal Disposal (Prime Minister's Office, Notice No. 40 of 1995). In vivo experiments were performed using seven-week-old or older male C57BL/6JJmsSlc mice or C57BL/6NCrSlc mice (SLC, Hamamatsu, Shizuoka, Japan). In vitro, experiments were performed using P12 male C57BL/6JJmsSlc mice purchased from SLC (Hamamatsu, Shizuoka, Japan). The mice for the in vitro experiments were housed under a 12-h dark-light cycle (light from 19:00 to 7:00) at $22 \pm 1\,°C$ with *ad libitum* access to food and water. For fiber photometry recordings, all animal procedures were conducted in accordance with the National Institutes of Health Guide for the Care and Use of Laboratory Animals and approved by the Animal Research Committee of Keio University School of Medicine (approval 14027-(1)). The mice for these experiments were housed individually and maintained on a 12-h dark-light cycle (light from 8:00 to 20:00). The body weights of the mice were maintained at 85% of their initial body weight. Water was readily available.

### Surgery

Surgery was performed to implant guide cannulas for pharmacological inactivation of the AIC, an optic fiber for optogenetic activation of the AIC, a microdrive with up to six electrodes for electrophysiological recordings, or an optic fiber for fiber photometry experiments. For fiber photometry experiments, mice were anesthetized with a ketamine-xylazine mixture (100 and 10 mg/kg, i.p.) Otherwise, they were anesthetized with isoflurane gas (1–2.5%).

For the implantation of guide cannulas for pharmacological inactivation of the AIC, an 8-mm midline incision was made from the area between the eyes to the cerebellum. Two craniotomies with a diameter of ~1 mm were performed using a high-speed drill (SD-102, Narishige, Setagaya, Tokyo, Japan) above the AIC (bregma: anterior-posterior (AP): 0.0 mm, medial-lateral (ML): ± 2.4 mm) or the posterior parietal cortex (PPC) (bregma: AP: −1.94 mm, ML: ± 1.5 mm). The dura was surgically removed. The guide cannulas were bilaterally inserted into the AIC (bregma: AP: 1.94 mm, ML: ± 2.4 mm: dorsal-ventral (DV): 3.0 mm) at 33.4 degrees or into the PPC (bregma: AP: −1.94 mm, ML: ± 1.5 mm: DV: 0.75 mm) vertically. Dummy cannulas were inserted into the guide cannulas.

For implantation of an optic fiber for optogenetic activation of the AIC, an 8-mm midline incision was made from the area between the eyes to the cerebellum. A craniotomy was performed at the following coordinates: 1.77 mm anterior to and 2.5 mm right of the bregma; the dura was surgically removed. A total volume of 500 nL (100 nL/min) of $AAV_{DJ}$-CaMKIIa-ChR2-EYFP ($1.93 \times 10^{13}$ vg/ml, Addgene #26969) was injected into the right AIC (AP: 1.77 mm, ML: 2.5 mm, DV 3.25 mm to the bregma) using a glass electrode, and an optical fiber (200 µm, 5 mm, Doric, Québec, Canada) was implanted 0.4 mm above the virus injection site and fixed with dental cement. For in vitro patch-clamp recordings, virus injection was performed in P13–14 mice. A 4-mm midline incision was made from the area between the eyes to the cerebellum. A craniotomy was performed at the following coordinates: 1.8 mm anterior to and 2.5 mm right of the bregma; the dura was surgically removed. A total volume of 500 nl (100 nl/min) of $AAV_{DJ}$-CaMKIIa-ChR2-EYFP was injected into the right AIC (bregma: AP: 1.77 mm,

ML: 2.5 mm, DV 1.65 mm) using a glass electrode, and the skin was sutured.

Electrode devices comprising a core body and a custom-made electrical interface board (EIB) accommodating up to 6 LFP channels, 2 electrocardiogram channels, 2 electromyography (EMG) channels, and 2 ground/reference channels were assembled for LFP and unit recordings as previously described[75]. The LFP electrodes were constructed from 17-μm-wide polyimide-coated platinum-iridium (90/ 10%) wire (California Fine Wire Co., Grover Beach, California, USA), and the electrode tips were plated with platinum to lower the impedance of the electrodes to 150–300 kΩ at 1 kHz. After anesthesia, an incision (~1 cm) was made on the upper chest, and 2 stainless-steel electro-cardiogram electrodes with a tip diameter of 147 μm (AS633; Cooner Wire Company, Chatsworth, California, USA) in which the polytetra-fluoroethylene (PTFE) coating at the tip (length: ~5.0 mm) was peeled off were sutured to the tissue underneath the skin of the upper chest. Then, the mouse was fixed in a stereotaxic instrument with two ear bars and a nose clamp. A midline incision was made from the area between the eyes to the incised neck area, and 2 stainless-steel EMG electrodes with a tip diameter of 147 μm (AS633; Cooner Wire Company) in which the PTFE coating at the tip (~5.0 mm) was peeled off were sutured to the dorsal neck muscles. Circular craniotomies with a diameter of ~1 cm were performed using a high-speed drill at coordinates of 2.4 mm right of the bregma for the AIC and 5.0 mm anterior and 0.5 mm bilateral to the bregma for the olfactory bulb (OB); then, the dura was surgically removed. The tips of the AIC tetrode bundles were lowered to the cortical surfaces at an angle of 33.4°, and the electrodes were inserted 0.5 mm into the brain at the final step of surgery. Stainless-steel screw-shaped electrodes were implanted on the surface of the craniotomies for the OB. In addition, stainless-steel screws were implanted on the surface of the cerebellum (9.6 mm posterior and 0.8–1.0 mm bilateral to the bregma) as ground/reference electrodes. All the wires and the electrode assembly were secured to the skull using dental cement.

For fiber photometry experiments, an 8-mm midline incision was made from the area between the eyes to the cerebellum. A craniotomy with a diameter of up to 1.5 mm was made at the coordinates of 1.1 mm anterior and 1.9 mm right or left of the bregma for the VLS using a high-speed drill, and the dura was surgically removed. A total volume of 500 nl GRAB$_{DA2m}$ virus (AAV-hSyn-GRAB$_{DA2m}$-W, a gift from Y. Li, $1.0 \times 10^{14}$ genome copies/ml) was injected through a pulled glass micropipette into the VLS (bregma: AP: 1.1 mm, ML: 1.9 mm, DV 3.5–3.7 mm) according to the atlas of Paxinos and Franklin (2004). The injection was driven at a flow rate of 100 nL/min by a Nanolitre 2020 Injector (World Precision Instruments, Sarasota, Florida, USA). The micropipette was left in place for another 5 min to allow for tissue diffusion before being retracted slowly. After GRAB$_{DA2m}$ virus injection, an optical fiber cannula (CFMC14L05; 400 mm in diameter, 0.39 numerical aperture; Thorlabs, Newton, New Jersey, USA) attached to a ceramic ferrule (CF440-10; Thorlabs) and a ferrule mating sleeve (ADAF1-5, Thorlabs) were inserted into the same side of the VLS as the virus injection and then cemented in place. The behavioral test and data collection were initiated more than 10 days after the surgery to allow the mice to recover and to allow the GRAB$_{DA2m}$ proteins to be expressed.

### Nose-poke behavior test

The nose-poke behavior test was conducted in a chamber made of acrylonitrile butadiene styrene polymer (300 mm in width × 200 mm in depth × 350 in height). The chamber was equipped with one or two air-puffing hole(s) with a diameter of 15 mm. When a mouse poked its nose into the hole, an air-puff stimulus (N$_2$ gas, 80 kPa, 100 ms) was given via a solenoid valve. In the two-hole chamber, one hole was associated with the air-puff stimulus, whereas the other was not. When a mouse continued poking its nose for more than 1 s, air puffs were administered every second. For the control condition, multiple toys, such as a ladder and a seesaw, were placed in the same chamber. After at least 3 days of handling, a mouse was placed in the enriched chamber for 15 min, allowed to habituate to the chamber and allowed to associate the air-puffing hole with the air-puff stimulus. Mice received air puffs within the first 1.7 ± 1.6 min (mean ± SD) of the habituation trial. The test trials started on the next day of the habituation trial. On each day, a mouse was placed alternately in each of the enriched and empty chambers twice. Each trial lasted 15 min, and the mouse was allowed to rest for 5 min in its home cage between trials. During the rest period, the chamber was cleaned with 70% ethanol. The test trials continued for at least 3 consecutive days.

### Passive avoidance test (step-through test)

The passive avoidance test was conducted while mice were in the apparatus (Model PA-3002; O'hara & Co., Ltd., Nakano, Tokyo, Japan), which consisted of a bright (~400 lux) compartment and a dark compartment. The two compartments were separated by a black partition with a semicircular doorway in the center. For the learning trial, a mouse was placed in the bright compartment, and the latency until it entered the dark compartment was recorded. When the mouse entered the dark compartment more than 5 cm from the partition, the reflective light sensor responded and applied an air-puff stimulus to the mouse's face. The mouse that received the air puff was immediately returned to the home cage. The test trials were performed 5 min later. The mouse was placed into the bright compartment again, and the time until it entered the dark compartment was recorded. The test trials were repeated 6 times at an intertrial interval of 5 min. When the mouse did not enter the dark compartment within 300 s, the trial was terminated, and the latency was recorded as 300 s.

### Open field test

In each experiment, a mouse was placed in the center of a square box made of acrylonitrile butadiene styrene polymer (300 mm in width × 300 mm in depth × 400 in height) with an open top. The tests were conducted in a room with a brightness of 150 lux. A camera was installed above the center of the field to monitor the instantaneous position of the mouse. The total distance traveled by each mouse for 10 min was measured.

### Pharmacological silencing

Immediately before drug injection, the dummy cannulas were removed, and stainless-steel cannulas were inserted (diameter: 0.37 mm; EICOM, Kyoto, Kyoto, Japan). As previously described[76], a volume of 0.3 μL Mus+Bac solution (0.05 μg/μL and 0.02 μg/μL, respectively) or saline was injected at a rate of 0.1 μL/min. Then, the injection cannulas were removed, and the dummy cannulas were again inserted at least one minute after injection. The behavioral test was initiated 15 min later. Each behavioral test was conducted every other day, and muscimol/baclofen solution or saline was injected alternatively before the test. To estimate the area of drug diffusion after the behavioral tests were complete, 0.3 μL of sulforhodamine 101 (50 μM) was injected at a rate of 0.1 μL/min, and mice were killed 15 min after the injection. The injected site was confirmed by confocal microscopy. For naltrexone administration, 10 mg/kg naltrexone hydrochloride (Abcam, Trumpington, Cambridge, UK) was intraperitoneally injected into the mice. Each behavioral test was conducted every other day, and naltrexone or saline was injected alternatively before the test.

### Slice preparation

Experiments were performed on four-week-old C57BL/6JJmsSlc mice in which ChR2 was expressed. Mice were anesthetized with isoflurane and decapitated, and their brains were removed. A coronal brain block (400 μm) containing the AIC was obliquely cut in the front-occipital direction using a vibratome (VT1200S; Leica, Shinjuku, Tokyo, Japan)

at a speed of 80 μm/s in ice-cold oxygenated modified artificial cerebrospinal fluid (222.1 mM sucrose, 27 mM NaHCO$_3$, 1.4 mM NaH$_2$PO$_4$, 2.5 mM KCl, 0.5 mM ascorbic acid, 1 mM CaCl$_2$, and 7 mM MgSO$_4$). Slices were then transferred to oxygenated artificial cerebrospinal fluid at 37 °C that was composed of the following: 127 mM NaCl, 3.5 mM KCl, 1.24 mM KH$_2$PO$_4$, 1.2 mM MgSO$_4$, 2.5 mM CaCl$_2$, 26 mM NaHCO$_3$, and 10 mM D-glucose; slices remained in the fluid for at least 30 min.

### In vitro patch-clamp recording

Experiments were performed in a submerged chamber perfused at 7–9 ml/min with oxygenated artificial cerebrospinal fluid at room temperature. Whole-cell patch-clamp recordings were obtained from small pyramidal neurons in the AIC, which were visually identified under an infrared differential interference contrast microscope. Patch pipettes (4–7 MΩ) were filled with a potassium-based solution consisting of (in mM) 135 potassium gluconate, 4 KCl, 10 HEPES, 10 creatine phosphate, 4 Mg-ATP, 0.3 Na2-GTP, and 0.3 EGTA. The signals were amplified and digitized at a sampling rate of 20 kHz using a MultiClamp 700B amplifier and a Digidata 1440 A digitizer that was controlled by pCLAMP 10.3 software (Molecular Devices). ChR2 was stimulated using a light-emitting diode (LED) (LEX2-LZ4; Brain Vision, Itabashi, Tokyo, Japan). The duration and intensity of the light pulses were 2.5 ms and 1.5 mW, respectively. The pulses were applied every 25 ms for 60 s. The 60-s stimulation period was repeated at intervals of 60 s for 15 min using a stimulator (SEN-3301; Nihon Kohden, Shinjuku, Tokyo, Japan).

### In vivo optogenetics

Mice expressing ChR2 were tethered to optic patch cords and connected to a laser source (473 nm, 100 mW, Lucir Inc., Tsukuba, Ibaraki, Japan) via a ferrule (Precision Fiber Products, Chula Vista, California, USA). The duration and intensity of the light pulses were 2.5 ms and 0.1–3.0 mW, respectively. The pulses were performed every 25 ms for 60 s.

### Adjusting the tetrode depth

Each mouse was connected to recording equipment via a digitally programmable amplifier (Cereplex M; Blackrock Microsystems, Salt Lake City, Utah, USA). The output of the headstage was entered into a data acquisition system (Cereplex Direct recording system; Blackrock Microsystems) via a lightweight multiwire tether and a commutator. The depth of the electrode tip was adjusted while the mouse rested in its home cage. Over at least 10 days following the implantation surgery, the electrode tip was advanced up to 31.25–250 μm per day until spiking cells were encountered in the AIC. The depth of the AIC was typically 2800–3500 μm. The tetrodes were then settled into the targeted area so that stable recordings could be obtained.

### In vivo electrophysiological recording

Electrophysiological recordings began after stable, well-separated unit activity was identified in the AIC. Data were sampled at 2 kHz and filtered between 0.1 and 500 Hz. Unit activity was amplified and processed with a 750 Hz high-pass filter. Spike waveforms above a trigger threshold of −60 μV were time-stamped and recorded at 30 kHz for 1.6 m. Recordings were conducted for at least 3 days.

### Fiber photometry and signal extraction

The changes in the extracellular concentration of dopamine were measured by a custom-made fiber photometric system (Olympus Corporation, Shinjuku, Tokyo, Japan)[37]. The fluorescence signal of the extracellular dopamine levels was obtained by stimulating GRAB$_{DA2m}$-expressing cells with a 465 nm LED (9.7 ± 0.1 μW at the fiber tip) and a 405 nm LED (the same power as that of the 465 nm LED). The 405 nm LED was used to correct movement artifacts. The 465 nm and 405 nm wavelengths were emitted by an LED alternately at 20 Hz (turned on

for 24 ms and off for 26 ms), with the timings controlled by a programmable pulse generator (Master-8; A.M.P.I., Jerusalem, Israel). Each excitation light was reflected by a dichroic mirror (DM455CFP; Olympus) coupled into an optical fiber cable (diameter: 400 μm, length: 2 meters, 0.39 NA, M79L01; Thorlabs) through a pinhole (diameter: 600 μm). The optical fiber cable was connected to the optical fiber cannula of the mice. The fluorescence signal was detected by a photomultiplier tube with a gallium arsenide phosphide photocathode (H10722–210; Hamamatsu Photonics, Hamamatsu, Shizuoka, Japan) at a wavelength of 525 nm. The fluorescence signal, transistor-transistor-logic signals for the times of LED excitations, and nose-poke signals from the behavioral apparatus were digitized by a data acquisition module (cDAQ-9178; National Instruments, Austin, Texas, USA). These digitized signals were simultaneously recorded at a sampling frequency of 1 kHz using a custom-made program (LabVIEW; National Instruments). The fluorescent signal was processed offline. Each 24-ms excitation period was identified according to the corresponding transistor-transistor-logic signal, and one averaged signal value was computed for each excitation period, with two frames excluded at both ends. Each value for 465 nm excitation was paired with a temporally adjacent counterpart for 405 nm excitation. The ratio of the 465-nm signal value to the 405 nm value yielded a processed signal that reflected the dopamine level in a ratiometric manner at a frame rate of 20 Hz.

### Histology

The mice implanted with guide cannulas were overdosed with isoflurane and perfused intracardially with phosphate-buffered saline (PBS) followed by 4% paraformaldehyde in PBS; then, the brains of the mice were removed. Brains were postfixed in 4% paraformaldehyde overnight, washed with PBS three times for 10 min each time, and sagittally sectioned at a thickness of 100 μm using a vibratome. The nuclei were counterstained with Hoechst 33342 (1:1000; ThermoFisher Scientific, Waltham, Massachusetts, USA). The mice implanted with optic fibers or electrodes were overdosed with isoflurane and perfused intracardially with PBS followed by 4% paraformaldehyde in PBS; then, the brains of the mice were removed. Brains were postfixed overnight in 4% paraformaldehyde and then equilibrated in 20% sucrose in PBS overnight followed by 30% sucrose in PBS overnight. Frozen coronal sections (50 μm) were cut using a microtome. ChR2-EYFP signals received no signal amplification, while GRAB$_{DA2m}$ was visualized with an anti-green fluorescent protein (GFP) primary antibody (1:500; ab13970; Abcam) and AlexaFluor-488 goat anti-chicken secondary antibody (1:500; A11039; Thermo Fisher Scientific). The nuclei were counterstained with Hoechst 33342 (1:1000). To check the positions of the electrode tips, serial sections were processed for cresyl violet and coverslipped. The positions of all electrodes were confirmed by identifying the corresponding electrode tracks in sections. Recordings were excluded from the data analysis unless the deepest position of the tetrode was located in the AIC. Fluorescence images were captured using an all-in-one microscope (BZ-X710; Keyence, Osaka, Osaka, Japan).

Following in vitro patch-clamp recordings, the electrodes were carefully withdrawn from recorded neurons. For visualization of patch-clamped neurons, the slices were immersed in 4% paraformaldehyde, followed by overnight postfixation. Then, the sections were incubated with 2 μg/ml streptavidin-Alexa Fluor 647 conjugate, blue fluorescent Nissl, and 0.3% Triton X-100 for 90 min. Fluorescent images were acquired using a confocal microscope (FV1000; Olympus) and were subsequently merged using ImageJ software.

### Data analysis

Nose-poke bursts were defined as follows. First, the nose-poke frequency was calculated by applying a Gaussian filter to the timing of nose pokes in one trial. Next, the poke frequency was calculated in the same way using poke-interval shuffled data. This shuffle calculation

was repeated 1,000 times, and the top 5% of the obtained values was set as the threshold. Nose-poke frequencies exceeding the threshold and with a poke interval of less than 1 s for three or more consecutive pokes were defined as burst events and omitted from the analyses, except for the analysis of intense behavioral states, as we describe hereafter.

We defined the onsets of the intense state as the timings at which mice performed nose pokes more than 2.4 times per minute. This definition was determined by the histogram of nose pokes (/min) of all mice: the nose-poke frequency approximated a distribution (Supplementary Fig. 14a) consisting of a Poisson distribution ($\lambda = 0.4$) and a log-normal distribution ($\mu = 2.1$, $\sigma = 2.0$), between which the frequency of nose pokes was minimized at 2.4 pokes/min. Therefore, mice that exhibited more than 2.4 pokes per minute were regarded as addicted. Note that the distribution of nose pokes was not derived from a normal distribution ($P < 0.001$, Lilliefors test).

LFP analysis was performed by applying a wavelet transform to the data and dividing by the total value of the entire frequency (1–200 Hz) to obtain normalized power. Electrocardiogram analysis was performed by applying a bandpass filter of 20–200 Hz and detecting the R-waves as maxima with values above a manually determined threshold at intervals of 50 ms or more. The noise was then manually detected and removed. HRVs (coefficients of variation in the RR interval[77]) were calculated every second from data that were downsampled to 400 Hz and smoothed with a 10-s moving average. EMG analysis was performed using data downsampled to 20 Hz of the signal, and the root mean square (rmsEMG) per second was calculated and used as power. For respiratory frequency, a wavelet transform was performed on the OB LFP waveform in the range of 1–10 Hz, and the frequency with the highest intensity was used as the respiratory frequency[25].

Spike sorting was performed offline using the graphical cluster-cutting software MClust[78]. Clustering was performed manually in two-dimensional projections of the multidimensional parameter space based on waveform amplitudes, the peak-to-trough amplitude differences, and waveform energies that were measured on the four channels of each tetrode. Cluster quality was measured by computing the $L_{ratio}$ and isolation distance[79]. A cluster was considered a cell when the $L_{ratio}$ was less than 0.30 and the isolation distance was more than 14, as previously described[80,81]. Autocorrelation and cross-correlation functions were used as additional separation criteria. Refractory periods of spikes were considered to increase confidence in the successful isolation of cells.

For cell responsiveness, we focused on the 10-s periods before and after nose pokes, and nose-poke responsive cells were defined as cells whose mean firing rates deviated from the 95% confidence interval of the shuffled data for more than 1 s. Among the responsive neurons, cells whose firing rates increased before poking were considered preexcited, cells whose firing rates increased after poking were considered postexcited, cells whose firing rates decreased before poking were considered preinhibited, and cells whose firing rates decreased after poking were considered postinhibited.

For analysis at the cell population level, time series of the average firing rates (bin = 2 s, calculated at a step of 0.5 s) for the 20-s periods before and after nose-poke events were concatenated for all cells (4–15 cells/day) recorded in a given day and subjected to principal component analysis. The resulting scores for principal components 1–3 in the 10 s before and after the poke were plotted (Supplementary Fig. 12b). We also performed principal component analysis on the data of the average firing rates of each cell during one 2-s period and another random 2-s period; in addition, we performed linear support vector machine analysis based on the obtained principal components 1 and 2 to classify whether the 2-s periods were related to nose pokes. The data were divided into 10 parts and cross-validated to obtain classification accuracy.

For the analysis of dopamine signals, the ratio of the values of excitation light at 465 nm to those at 405 nm was adopted. For the fluorescence intensity ratio at a given time point, the median ± 10% mean ($R_0$) value for the 2.5-s periods before and after was subtracted and divided by $R_0$ to obtain the fluorescence intensity change ratio $\Delta R/R_0$. The difference from the baseline was quantified using the following time points: 4 s to 2 s before the nose pokes (or other stimulus presentation) as a baseline period, 1.5 s to 0 s before nose pokes as the prepoke period, 0 s to 1 s after nose pokes as the during-poke period, and 1 s to 2.5 s after the nose pokes as a postpoke period.

All statistics that compared the difference between one sample and a certain value or between two samples were derived by a bootstrap test as previously described[82]. When the data were paired (e.g., empty and enriched trials), bootstrap confidence intervals were calculated for the difference (or ratio) of the compared data, and a significant difference was considered to exist when the upper limit of the 95% confidence interval was negative or the lower limit was positive. A ratio was significantly higher or lower than 1 when the upper limit of the 95% confidence interval was less than 1 or when the lower limit was greater than 1, respectively. In the case of unpaired data, bootstrap confidence intervals were obtained from the mixed data of the two groups, and the same test was performed. Bonferroni correction was used for multiple comparisons. For correlation plots, significance was determined based on Pearson's correlation coefficient. The Jonckheere trend test was used to evaluate increasing and decreasing trends in the data. The chi-square test was used to test for differences in proportions.

### Reporting summary
Further information on research design is available in the Nature Portfolio Reporting Summary linked to this article.

## Data availability
All raw data obtained in this study are kept strictly within the laboratories of the University of Tokyo in accordance with the guidelines. The data that support the findings of this study are available from the corresponding author upon a request. Source data are provided with this paper.

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

## Acknowledgements

This work was supported by JST ERATO (JPMJER1801), the Institute for AI and Beyond of the University of Tokyo, and JSPS Grants-in-Aid for Scientific Research (18H05525).

## Author contributions

Y.Y. and Y.I. conceptualized the study; Y.Y., Y.S., O.J., K.M., T.K., K.I., and A.Y. performed the experiments; S.M. procured the ChR2 virus; S.Y. produced the GRABDA2m virus; Y.Y. performed the data analysis; Y.Y. and Y.I. wrote the original draft; Y.Y., Y.S., K.M., T.K., K.I., A.Y., S.M., S.Y., K.F.T., and Y.I. reviewed and edited the final manuscript.

## Competing interests

The authors declare no competing interests.
