## [Peer Review File · Nature Communications]

Mesolimbic dopamine release precedes actively sought aversive stimuliREVIEWER COMMENTS

Reviewer #1 (Remarks to the Author):

The authors put mice in empty or enriched chambers within which the mice could push a button that would deliver an aversive stimulus as a model (based on Wilson's "Just think" paper) of boredom. Animals in the empty room made more nose pokes to receive the aversive stimulus and aversive stimuli led to an increase in DA. The authors highlight the insula as the site for the boredom avoidance behaviours. Furthermore, some animals became addicted to the aversive stimulus and opioid blockers eliminated the transition to addictive behaviours.

The paper is comprehensive and provides an interesting and potentially valid animal model of boredom. I have only some minor points to make, but should highlight that my expertise only really allows me to focus on the observed behaviours. I can not evaluate the physiological and optical imaging aspects of the work as this is outside my area of expertise.

Lines 109 – 111 starts with “In humans,..” but then references mainly work in rats with respect to arousal and boredom. In humans the story of arousal and boredom is complex – some say it is low arousal (Barmack, 1939; Van Tilburg & Igou, 2011 - via self report; Vogel-Walcutt et al., 2012) and others high arousal (Berlyne, 1960; Jang et al., 2015; Merrifield & Danckert, 2014; note this is not an exhaustive reference list and the debate splits about 50/50 for boredom as a high/low arousal state). One coherent argument (Elpidorou, 2021) suggests that arousal should not even be considered part of the definition of the state. In addition, Hoemann et al., 2020 highlight that psychophysical changes are generally non-specific to affective states. When moving to an animal model it then becomes difficult to evaluate the psychophysical changes. I don't think the authors can resolve this, but it would be good to acknowledge the challenge - both definitional (is boredom high/low/mixed arousal) and interpretive with respect to animal models.

In a similar vein, how do the authors resolve the discrepancy between work suggesting that HRV decreases under stressful circumstances, but was increased after nose pokes here? Boredom is aversive and so might be expected to show decreased HRV. Can the authors explore HRV just prior to nose pokes? Is this post nose-poke data a release of stress?

What are we to make of roughly equivalent populations of AIC neurons increasing and decreasing firing rates around a nose-poke event? Doesn't this make the association ambiguous?

Figure 1d – the nose pokes are substantially higher (regardless of condition) in the without air puffs condition – so overall should we see the effect of the air puffs as decreasing exploratory behaviour in the animals?

Minor things:

Line 146 “resulting a decrease in...” is missing “in” prior to “a decrease”

Line 235 – is “predicted” meant to be “preaddicted”?

Reviewer #2 (Remarks to the Author):

In this paper, the authors tried to identify the putative mechanism for acquiring addictions. To do so, they proposed a simplistic method to assess behavioral addiction, employing two boxes, one empty and the other with toys. In the boxes, there was a nose-poke hole in its wall by which they deliver air-puff that has been considered aversive. They found that activation and inhibition of the firing patterns of insular cortex neurons could increase and decrease the frequencies of air-puff self-stimulation. In addition, they determined that dopamine concentrations in the ventrolateral striatum increased when the animals actively sought air puffs applied to their faces, like the reported expected food rewards.

Interestingly, they found that some animals in an empty box repeatedly developed self-stimulation addiction-like behavior that an opioid receptor antagonist, naltrexone, could reverse. The authors concluded that being in an empty box induced a boredom condition that triggered a transition to behavioral addiction, measured by increased frequency of receiving nose pock air-puffs. These findings could underpin the neural mechanisms of addiction and personality disorders.

Although the findings are interesting and potentially significant, several issues should be solved before the paper can further be considered for publication in Nature Communications.

The first criticism would be the term boredom-like behavior in mice. Unfortunately, this term is ambiguous and anthropomorphized and does not allow us to understand the phenomenon they describe. I suggest changing the term impoverished environment in the title and the paper. Other

factors have not been evaluated: they do not describe whether the phenomenon of increased air puffs depends on the passage of time or whether it is attributable to novelty, nor do they discuss whether there is a learning process.

In the results section, P4, L 90-93, they stated..." that the frequencies of nose pokes were, on average 2.1 times higher in the empty sessions, "... suggesting that mice were more likely to be bored in the empty chamber"... The increase in behavior cannot be attributed to boredom, especially if they have not given an operational definition in the introduction. The only close description will be that animals' in-home boxes tend to be more approximative to various stimuli than animals in enriched homes. I think the authors should constrain their interpretations to more operationally measures of the mice behaviors.

Similarly, on P4, L 113-122, the authors indicate that after the presentations of the air-puff, there is a decrease in heart rate and an increase in heart rate variability. However, it is necessary to show that changing mice from an enriched box to a depleted box increases HR and decreases HRV, showing that environment changes promote these biometric data modifications.

On P5, L134, the authors stated that "...also reduced locomotor activity in the test chamber". However, the title of Supplementary Figure 4 says that "GABA receptor blockade of the AIC does not alter locomotor activity." There are discrepancies since the results show a decrease in the locomotor response. Furthermore, it should be clarified that both muscimol and baclofen are GABA receptor agonists.

They also stated in P5, L142, "...nose pokes reduced by AIC deactivation cannot be explained by the reduced locomotion alone,"... but neither do they suggest nor discuss why it is due, and with that, they assume that it is the center of "boredom."

It is important to clarify when the Gamma oscillations measurements occurred and if they are time-dependent. If the increase in gamma in IC is the "neural representation of boredom," this should increase or be sustained over time, and they do not mention/discuss this possibility. In addition, they indicate that "...increasing the AIC gamma oscillations may cause an increase in nose pokes," but they do not report whether this phenomenon is consistent in each nose poke that the animals executed or if there is a change in the first nose pokes versus the last due to learning process. It would be worth plotting Empty 1 vs. Empty 2 boxes both in behavior and records. At this point and the next, it is necessary to show whether these results are also observed in enriched environments to attribute them to boredom.

It would be necessary to rule out that the optogenetic stimulation of the AIC increases nose pokes in the enriched environment; this could strengthen the idea that the activity of the IC only increases in

impoverished environments. They should also show if the optogenetic stimulation coincides with the nose poke execution since they only show the session average.

On P7, L207-9, they stated, "...when mice actively performed nose pokes to receive air puffs, dopamine levels increased before and during nose pokes..." This statement implies a learning process because the animals are expecting something. However, it is not indicated at what time the training measurement occurred.

It is unclear why they decided to try an opioid antagonist instead of a dopaminergic one if they wanted to demonstrate a dopaminergic effect?

In the discussion section, the authors mention that "they show that the insular cortex is the regulatory center for 'boredom-avoidance' behavior in mice." However, the evidence presented is not enough to ensure this statement.

Reviewer #3 (Remarks to the Author):

In this study, Yawata and colleagues establish a creative behavioral paradigm in which mice are more likely to nose-poke for air puffs when they are in an empty cage than in an enriched environment. The authors further explore this behavior by simultaneously recording physiological variables and dopamine release in the ventral striatum. By applying optogenetic tools to study the insular cortex, the authors were able to bi-directionally influence the extent to which mice were willing to self-administer air puffs, implicating this region of cortex in these behaviors. The authors further claim that a subset of mice progress to high rates of air puff self-administration that is no longer decreased by environmental enrichment but that such progression to high rates of self-stimulation could be prevented via pre-treatment with the opioid receptor antagonist naltrexone.

Overall, this work describes a novel behavioral paradigm that reveals interesting context-dependence and inter-subject heterogeneity in how mice engage in exploratory and information-seeking behaviors despite potentially aversive consequences. However, framing this behavior as revealing "boredom" and

describing the high-self-administration subpopulation of mice as “addicted” to the air puff stimulation constitute over-reaching interpretations of these results. Furthermore, the use of these terms have specific connotations and definitions in human behavioral studies, thereby risking confusing potential readers from these fields. Moreover, these strong interpretations should be weighed carefully against other equally or more compelling alternative interpretations that these behaviors reflect drives for prioritized information seeking or exploration of the most salient stimuli currently available in the environment.

This work contains rich and rigorous behavioral characterization that could be of interest to the readership of Nature Communications. However, the current framing and discussion of these findings around boredom and addiction is overly strong and not compelling in its current form. If the authors reconsider the use of these interpretations and address the other technical concerns presented below, this work would be of interest to experts in neuroethology as well as to many neuroscientists with interests pertaining to the function of the insular cortex and mesolimbic dopamine system.

Major comments:

1) Regarding the interpretation that the behavioral effects reflect boredom: First, just because the airpuff is aversive when it is passively applied in an unexpected fashion, this doesn't mean it is necessarily aversive when expected during self-stimulation. Many strong stimuli can be aversive when unexpected, likely due to sensory prediction errors. Relatedly, animals may change their motor behavior to mitigate the aversive aspects of the stimulus. For example, during cued airpuff to the face, rodents learn to close their eyes ahead of the airpuff.

2) Locomotion controls (Supp. Figures SF 4 and SF 8): These figures further illustrate why this behavioral paradigm cannot be considered to only represent “boredom” behaviors. Differences in information seeking, exploratory drives, or decreased perceived aversiveness of the self-administered air puffs may account for some of these behavioral effects. A decrease in movement is very likely to cause a decrease in exploration of every part of the arena. In complete learning coupled to this change in exploration could also cause such exploration to modify the rate of behaviors that promote repeated airpuff self-stimulation, at least in part.

- SF 4 – This figure, titled “GABA receptor blockade of the AIC does not alter locomotor activity,” is misleading. Muc + bac in AIC does indeed affect locomotion variables presented in panels A and C. Further, these effects may be stronger in empty environments versus enriched ones, but the authors do not fully discuss these nuances. It is also unclear as to why muc +bac in AIC changes the distance traveled in the empty/enriched task but not in the OFT results presented in panel E.

- SF 8 – Same issue as SF4 – naltrexone does have effects on locomotion – which again may be stronger in the empty condition. Again – it is confusing that there are no effects of naltrexone on OFT activity.

3) Fig 1 + 2: Fig 1 and 2 could be combined to more clearly present the behavioral paradigm to readers.

- The experimental design includes 4 blocks, but there is no control for behavioral habituation/acclimation to the potential neophobia associated with the airpuff. The four sequential conditions (enriched-1, empty-1, enriched-2, empty-2) are not counterbalanced, and are grouped into two bars (all enriched, or all empty-cage). Might the ordering drive part of the observed effect? This could be examined by showing the mean for all 4 blocks in the bar graphs in Fig1C,D,E and Fig 2C. As it will allow estimation and ruling out of whether e.g. enriched-2 also shows more self-stimulation than enriched-1 due to a gradual upward trend. Most convincing would be if empty-1 showing more self-stimulation than both enriched-1 and enriched-2 when considered separately.

o If the above requested analyses call the results into question, a more compelling experiment would be to demonstrate that the behavioral effects remain the same if the exposure to the box conditions is reversed (e.g. – empty – enriched – empty – enriched)

- There is significant discrepancy in the sample sizes between conditions in these two figures, that should be addressed and / or justified via a power analysis of expected effect sizes (e.g., Fig 1, n = 4 in “no air puff” vs. n = 23 in “air puff” conditions).

- The use of bootstrapping is useful given later claims about non-normal distribution of nose poke behaviors.

4) Fig 3 – nose poking necessitates a stop in locomotion. How much of a drop in HR and increase in HRV would you expect simply from this behavioral change? The ~10 bpm change observed around the nose poke entry appears to be a small effect size. In general, this figure distracts from the rest of the work and may be better suited to be a supplementary figure.

5) Fig 4 – Panels A,B,F,G seem like useful loss and gain of function experiments, pending interpretational issues related to motivational drives. The manipulation of activity in anterior insular cortex appears to be somewhat similar to manipulations of anterior ‘sweet taste cortex’ by Charles Zuker’s group.

o Jin, H., Fishman, Z.H., Ye, M., Wang, L., and Zuker, C.S. (2021). Top-Down Control of Sweet and Bitter Taste in the Mammalian Brain. *Cell* 184, 257–271.e16. <https://doi.org/10.1016/j.cell.2020.12.014>.

o Wang, L., Gillis-Smith, S., Peng, Y., Zhang, J., Chen, X., Salzman, C.D., Ryba, N.J.P., and Zuker, C.S. (2018). The coding of valence and identity in the mammalian taste system. *Nature* 558, 127–131. <https://doi.org/10.1038/s41586-018-0165-4>

For example, stimulation of this region by the Zuker group has promoted seeking behaviors for aversive stimuli, which seems highly relevant here, and may also modify the interpretation for the role of AIC as a general driver of seeking behaviors in more complex contexts with competing behavioral choices.

The title of this figure appears totally different from the above loss/gain or function: “Gamma oscillations in the AIC increase before nose-poking behavior,” and should be modified. More generally, the discussion focusing on gamma oscillations (panels C-E) in general appears somewhat arbitrary and distracts from the main messages of this paper. In this reviewer’s opinion, it might be best to remove these panels from the paper entirely, or at the least to move to the supplement. Interpretation of such changes in rhythmicity is challenging as it’s likely to occur in many other brain regions as well, and to be related at least in large part to arousal and motor behaviors rather than strictly to learning or behavioral choice.

- In Fig 4D, please include additional spike rasters (as in SF6C)

- Are all opto / pharmacological AIC manipulations done in “empty box” conditions?

6) Fig 5 – As with Figure 4C-E, it is unclear what Figure 5 really adds to the work. Instead, it seems to distract from the main messages. Overall the yield in neurons (N=93) is quite low and the insights gained appear minimal. The decoding accuracy is not that impressive, and it is unclear if neural recordings in AIC are special in any way or if similar results would be seen in most brain areas. Perhaps this figure could be removed or these data could be moved to the supplement prior to Figure 4, to motivate somewhat the use of silencing of insular cortex?

- To claim that the AIC is uniquely involved in this behavior, it would be more compelling to compare AIC activity across nose pokes that are paired with an air puff versus nose pokes that are not paired with an air puff, and to also record in other brain areas, ideally at the same time.

7) Fig 6 –

- The authors claim in the subheading for the manuscript that “Aversive stimuli act as a reward when mice were bored.” Here, they claim that “anticipation of the air puff becomes a reward”. Neither interpretation is entirely compelling. The discussion around salience vs. valence encoding of dopamine release in the VLS is confusing and not entirely consistent with existing literature – e.g. - de Jong et al. 2018 (cited in line 203). As the authors mention, some of the dopamine release may reflect salience rather than reward or reinforcement (see above).

o Also, it is not well substantiated that dopamine release in the VLS reflects “how mice perceive” events (line 198).

8) Fig 7 – There are concerns with using the term “addiction”.

- This is not a conventional use of the term. Also, the authors appear to define the 'ramping up' of self-stimulation compared to a very short initial period of measurement, It's unclear if this is just the animal getting over neophobia of some sort. For example, would this development of "addiction" behavior still occur in the empty context if the mice were thoroughly habituated to the airpuff self-stimulation in the enriched context?

- It is not convincing that the dopamine transference to the pre-air-puff time in the "post addiction" state is a feature of addiction versus simply habituation and learning from repeated presentations. Should show comparably timed dopamine recordings from "non-addicted mice"

- Pre-treating mice with naltrexone before an exposure to the behavioral paradigm is not similar to how this drug is used in humans, where it is used to treat substance use disorders once they have developed. The clinical relevance is thus not clear. Further, the effects are also consistent with a decreased exploratory drive. Have the authors then tested the same mice without naltrexone to see if the addictive behaviors develop?

- A more conventional application of naltrexone might be to examine the effects of naltrexone on the behavior and / or ventral striatum dopamine dynamics in mice that are already displaying high levels of air puff self-administration.

Minor:

1) It would be useful to put this study in the context of literature in which neuroscientists have described prioritized salience maps of the environment and how these guide exploration (e.g., in parietal cortex). Further, it would be useful to compare the current work to studies of animal preferences for medium-preference fruits in primates depending on whether higher or lower preference fruits are available, corrects in dopamine neuron activity. Might these studies have parallels to the current findings?

2) Fig 6 –

- Incorrect use of the term "imaged" when referring to photometry recordings

- Are the reported light power values of 10 uW for the 465 and 405 LED correct? This seems low and is perhaps may be due to the authors not accounting for the LED duty cycle? Does this represent peak power or average power?

- B: food pellet positive control – please clarify if mice were food restricted

o Please add error bars for trace in bottom left

- C,D,E: are these all recorded in the empty cage condition?

o D: Please add error bars for traces?

-

3) SF 3 – The histology summary for Fig 4 manipulations is nice. However, comparing sagittal and coronal placement for different manipulations is difficult for the reader.

4) SF 7 – The claim that the population distribution of nose poke behavior is bimodal is not compelling based only on the histogram in panel A – this be more compelling to demonstrate this via a statistical test for bimodality. It is unclear that this point is worth pushing too strongly.

5) Please add error bars in Figure 6E and 7E

6) Please add Deng et al., 2021 Cell citation for insular cortex stimulation ☐ VLS dopamine discussion, line 291

Manuscript number: NCOMMS-22-22844-T

Yawata et al., "Mesolimbic dopamine release precedes actively sought aversive stimuli"

January 4th, 2023

Comments and Answers are below. The main revised sections are shown in **dark red** throughout the manuscript.

Reviewer #1

Thank you for the positive evaluations, which have encouraged us to resubmit this manuscript. We have revised our manuscript in accordance with your comments. Our point-by-point responses are as follows:

1-1) Lines 109 – 111 starts with “In humans,..” but then references mainly work in rats with respect to arousal and boredom. In humans the story of arousal and boredom is complex – some say it is low arousal (Barmack, 1939; Van Tilburg & Igou, 2011 - via self report; Vogel-Walcutt et al., 2012) and others high arousal (Berlyne, 1960; Jang et al., 2015; Merrifield & Danckert, 2014; note this is not an exhaustive reference list and the debate splits about 50/50 for boredom as a high/low arousal state). One coherent argument (Elpidorou, 2021) suggests that arousal should not even be considered part of the definition of the state. In addition, Hoemann et al., 2020 highlight that psychophysical changes are generally non-specific to affective states. When moving to an animal model it then becomes difficult to evaluate the psychophysical changes. I don't think the authors can resolve this, but it would be good to acknowledge the challenge - both definitional (is boredom high/low/mixed arousal) and interpretive with respect to animal models.

We thank this reviewer for the very valuable comments and insights. As the reviewer states, boredom is a subjective feeling and has historically been considered in terms of human psychology and physiology. The views on boredom are not uniform among researchers, and boredom is not monophasic but has several different aspects, including aspects that may seem somewhat contradictory. Against this confusing background, we attempted to construct an animal model to quantitatively assess "boredom". However, since the definition of boredom is ambiguous, it is logically impossible to argue whether the aversion-avoidance behavior exhibited by our rats reflected a "bored" psychological state. In response to the other reviewers' comments and the editor's suggestions, we decided to focus only on the actual facts observed during this study and to separate this study from the context of boredom or addiction. We believe that this major contextual revision has reduced the extravagance of the paper and has resulted in a paper with a clear skeleton. Moreover, we found that the value of our findings did not diminish in this new context and that the findings continue to provide important novel insights into the relationship between actively sought aversive stimuli and mesolimbic dopamine release. We deeply appreciate the opportunity to make this fair and scientific revision.

1-2) In a similar vein, how do the authors resolve the discrepancy between work suggesting that HRV decreases under stressful circumstances, but was increased after nose pokes here? Boredom is aversive and so might be expected to show decreased HRV. Can the authors explore HRV just prior to nose pokes? Is this post nose-poke data a release of stress?

This comment is rooted in keen insight. We think the same way; that is, it is consistent to interpret the change in these physiological parameters after nose pokes as a release from a state

of stress. As the reviewer suggested, we measured the 'baseline' HRV before nose pokes and found that the HRV was higher in the empty chamber than in the enriched chamber. This result also confirms our idea that vacant rooms are stressful. The data are shown in Supplementary Figure 6c.

1-3) What are we to make of roughly equivalent populations of AIC neurons increasing and decreasing firing rates around a nose-poke event? Doesn't this make the association ambiguous?

The simultaneous increase and decrease in firing is a widespread property of neural circuit responses (for example, *Curr Opin Neurobiol*, 7:514-522, 1997). We believe that these responses probably enhance the signal-to-noise ratio by increasing the contrast of the signal (e.g., via lateral inhibition or balanced E/I). This has been described in the legend of Supplementary Figure 11d.

1-4) Figure 1d – the nose pokes are substantially higher (regardless of condition) in the without air puffs condition – so overall should we see the effect of the air puffs as decreasing exploratory behaviour in the animals?

Yes, it was possible that the air puffs reduced exploratory behavior, but here, we believe that the number of pokes decreased because the air puff *per se* was an aversive stimulus for the mice. In fact, we have performed a new data analysis and found that the amount of locomotor activity itself did not decrease in the chamber with air puffs; this finding was added to the revised manuscript. Since mice have an instinctive habit of poking their noses in a hole in the chamber wall, we speculate that poking increased in the absence of air puffs. We think that the data could be additional evidence that an air puff is essentially an aversive stimulus. The data are shown in Supplementary Figure 3.

1-5) Line 146 "resulting a decrease in..." is missing "in" prior to "a decrease"

Thank you for pointing out our typo. We have corrected it.

1-6) Line 235 – is "predicted" meant to be "preaddicted"?

Thank you for pointing this out. In response to the other reviewer, we have avoided using the word "addiction" as described below.

Reviewer #2

We are grateful to this reviewer for her or his constructive comments, which have greatly improved our work. Individual responses are provided below.

2-1) The first criticism would be the term boredom-like behavior in mice. Unfortunately, this term is ambiguous and anthropomorphized and does not allow us to understand the phenomenon they describe. I suggest changing the term impoverished environment in the title and the paper. Other factors have not been evaluated: they do not describe whether the phenomenon of increased air puffs depends on the passage

of time or whether it is attributable to novelty, nor do they discuss whether there is a learning process.

Thank you for the important comments. As this reviewer says, there are diverse aspects in "boredom". We were also not sure if the mice truly felt subjective boredom. We have taken into account the comments of the other reviewers and have decided not to use the word "boredom" in this revised manuscript. Instead, we have tried to describe the paper in a direct style, purely explaining the facts without any ambiguous interpretations.

In accordance with the latter comment, we also performed a new analysis of the time dependence of poke frequency, confirming a gradual increase in poke frequency over the course of the 15-minute trial. This trend was statistically significant in the trend test, and these findings were added to the revised manuscript. We cannot completely rule out the possibility that novelty and learning processes are involved in our data, but their effects are small, given that the animals were habituated to the chamber the day before the test and that the behavioral test was conducted over a total of three days. The data are shown in Supplementary Figure 2.

2-2) In the results section, P4, L 90-93, they stated..." that the frequencies of nose pokes were, on average 2.1 times higher in the empty sessions, "... "suggesting that mice were more likely to be bored in the empty chamber"... The increase in behavior cannot be attributed to boredom, especially if they have not given an operational definition in the introduction. The only close description will be that animals' in-home boxes tend to be more approximative to various stimuli than animals in enriched homes. I think the authors should constrain their interpretations to more operationally measures of the mice behaviors.

This is a critical comment on the main focus of this paper. We also deeply agree with this comment. Based purely on the behavior exhibited by our mice, we cannot identify what we observed as being due to boredom and must consider many other possibilities. Therefore, in the revised manuscript, we have made every effort to convey the observed facts as they are without inflating or trivializing them. We are confident that the experimental data we have obtained are of sufficient scientific value, even without the context of boredom.

2-3) Similarly, on P4, L 113-122, the authors indicate that after the presentations of the air-puff, there is a decrease in heart rate and an increase in heart rate variability. However, it is necessary to show that changing mice from an enriched box to a depleted box increases HR and decreases HRV, showing that environment changes promote these biometric data modifications.

Thank you for this comment. We agree with this reviewer that such analysis can yield deeper insight. However, in practical terms, the situation was slightly more complex. In fact, heart rates tended to be higher in the enriched environment than in the empty environment. This may be because mice played with toys in the enriched chamber (that is, there was a difference in physical activity). Please see the figure below. This is why we did not compare the values between environments but compared values before and after nose pokes. Instead, we also compared the mean heart rate and its variability between the empty chamber and the home cage, in which mice exhibited similar levels of locomotor activity. Data are now shown in Supplementary Fig. 5, indicating that the HR and HRVs in the empty chamber are higher and lower, respectively, than those in the home cage.

2-4) On P5, L134, the authors stated that "...also reduced locomotor activity in the test chamber". However, the title of Supplementary Figure 4 says that "GABA receptor blockade of the AIC does not alter locomotor activity." There are discrepancies since the results show a decrease in the locomotor response. Furthermore, it should be clarified that both muscimol and baclofen are GABA receptor agonists.

We have changed the figure title to "Infusion of GABA receptor agonists into the AIC does not alter locomotor activity in novel open fields", which is the main point we would like to state here. Additionally, thank you for raising the question about the actions of muscimol and baclofen. Indeed, they are agonists, not antagonists.

2-5) They also stated in P5, L142, "...nose pokes reduced by AIC deactivation cannot be explained by the reduced locomotion alone,"... but neither do they suggest nor discuss why it is due, and with that, they assume that it is the center of "boredom."

Thank you for this comment. While we have already removed the context of boredom from the paper, this comment remains important. What we would like to say was that the decrease in locomotion is the only cause of the AIC inhibition-induced decrease in nose pokes, since the open field test failed to show a decrease in locomotion. Moreover, not only the numbers of nose pokes per time but also the numbers of nose pokes per unit of distance traveled were reduced by AIC inhibition. Although AIC inhibition could reduce motivation or locomotion, the effects were still confirmed after the corrections of these factors. Thus, we believe that AIC inhibition attenuated the increase in nose pokes in an empty environment. This line of argument was not carefully explained in the original manuscript. In the revised manuscript, this point has been reiterated and explained in a step-by-step manner (P5, L33).

2-6) It is important to clarify when the Gamma oscillations measurements occurred and if they are time-dependent. If the increase in gamma in IC is the "neural representation of boredom," this should increase or be sustained over time, and they do not mention/discuss this possibility. In addition, they indicate that "...increasing the AIC gamma oscillations may cause an increase in nose pokes," but they do not report whether this phenomenon is consistent in each nose poke that the animals executed or if there is a change in the first nose pokes versus the last due to learning process. It would be worth plotting Empty 1 vs. Empty 2 boxes both in behavior and records. At this point and the next, it is necessary to show whether these results are also observed in enriched environments to attribute them to boredom.

Thank you for pointing us in a new direction of analysis. Following this comment, we first examined the time evolution of gamma oscillations. The gamma power changes are plotted separately for the first 1/4 and the last 1/4 nose-poke events, indicating that the increase in

gamma oscillations was more pronounced in the last 1/4 nose pokes. Data are shown in Supplementary Fig. 9a. In addition, the gamma oscillations in the enriched sessions were higher than those in the empty sessions and increased toward nose pokes. This gamma power increase in the enriched sessions might reflect the motivation to seek aversive stimuli in spite of the presence of the toys. In the revised manuscript, this point has been explained on P6 L24 and in Supplementary Fig. 9b.

2-7) It would be necessary to rule out that the optogenetic stimulation of the AIC increases nose pokes in the enriched environment; this could strengthen the idea that the activity of the IC only increases in impoverished environments. They should also show if the optogenetic stimulation coincides with the nose poke execution since they only show the session average.

We previously conducted this analysis. Unlike in the empty chamber, the same optogenetic stimulation of the AIC did not increase nose pokes in the enriched chamber. The data were not shown in the previous manuscript, but they are now presented in Supplementary Figure 10f. We speculate that even if "boredom" is artificially augmented in the enriched chamber, it is mitigated by playing with toys, so there is no need to poke. Additionally, the effect of optogenetic stimulation did not coincide with nose poke execution. This is perhaps because AIC activation changed the boredom-like 'feeling' and thus had long-lasting effects on the behaviors of mice, but it did not directly 'trigger' nose-poking behaviors. Therefore, we showed the data as the session average.

We have also created two peri-stimulus time histograms (PSTHs) showing the change in frequency when the onset and offset of opto-stimulus timing were aligned to Supplementary Fig. 10d). These data showed a specific change in the frequencies of nose pokes, which is again consistent with the idea that AIC stimulation does not directly initiate nose poking.

2-8) On P7, L207-9, they stated, "...when mice actively performed nose pokes to receive air puffs, dopamine levels increased before and during nose pokes..." This statement implies a learning process because the animals are expecting something. However, it is not indicated at what time the training measurement occurred.

As this reviewer pointed out, we did not describe our schedule of experiments in detail. The day before we started the experiments, we placed the mice in the same chamber and demonstrated to them again that they would receive air puffs as they performed nose pokes. In other words, we conducted the experiments with mice that had already learned about the tricks of the chamber, and thus, the prior experimental procedure may have some effect on the dopamine data. To help readers gain a more precise understanding, we have specified the experimental procedure in the Methods section (P21, L31).

2-9) It is unclear why they decided to try an opioid antagonist instead of a dopaminergic one if they wanted to demonstrate a dopaminergic effect?

This is partly because naltrexone is already in clinical use for the treatment of addiction but also because our preliminary experiments showed that naltrexone suppressed the dopamine increase immediately before nose-pokes. From this fact, we hypothesized that naltrexone might inhibit the transition to the addiction state. Upon receiving this comment, we thought that the dopamine data might be important in our motivation for naltrexone use and have included them in this revised manuscript (Supplementary Fig. 14). The idea of dopamine antagonists is also important, and indeed, we tried some experiments with these drugs, but the experiments were not successful because the antagonists greatly reduced the motivation/locomotion of the mice.

Incidentally, we have also decided not to use the term ‘addiction’ in this manuscript because, like ‘boredom’, it was not strictly defined and raised concerns about our argument in the previous manuscript, and we have now simply referred to it as an "intense state" with high nose-poke frequencies. These changes in careful writing were made possible largely due to the comments of this reviewer, and we would like to express our gratitude once again.

2-10) In the discussion section, the authors mention that "they show that the insular cortex is the regulatory center for 'boredom-avoidance' behavior in mice." However, the evidence presented is not enough to ensure this statement.

Thank you for pointing this out. The word "center" was clearly an overstatement and has been removed from the revised manuscript.

Reviewer #3

We appreciate that this reviewer raised several important issues. We are pleased to have been able to revise the paper so that it is a better manuscript based on these comments. Individual responses are listed below:

3-1) Regarding the interpretation that the behavioral effects reflect boredom: First, just because the airpuff is aversive when it is passively applied in an unexpected fashion, this doesn't mean it is necessarily aversive when expected during self-stimulation. Many strong stimuli can be aversive when unexpected, likely due to sensory prediction errors. Relatedly, animals may change their motor behavior to mitigate the aversive aspects of the stimulus. For example, during cued airpuff to the face, rodents learn to close their eyes ahead of the airpuff.

Thank you for your insightful comments. The day before we started the experiments, we placed mice in the same chamber, and thus, the mice had already learned about the tricks of the chamber in which they would receive air puffs as they performed nose pokes. In our experimental conditions, therefore, an air puff was not an ‘unexpected’ stimulus for mice, except for the “passive air puff” conditions in Figure 4d, e. Moreover, we summarized the data from all nose-poke trials (not only from the first trials of each mouse). Despite these conditions, a decrease in dopamine concentration was observed after receiving air puffs, suggesting that air puffs functioned as an aversive stimulus. This idea is consistent with the observation of air puff avoidance in the step-through test (Supplementary Fig. 1) and the increase in nose-poke in the "w/o air puff" conditions (Fig. 1d). We did not conduct eye-blink conditioning, but our previous study also showed that rodents hesitated (avoided or slowed down) to enter an area with air puffs when they were allowed to freely explore a space (Okada et al., *Front. Neural Circuits*, 11:101, 2017).

3-2-1) Locomotion controls (Supp. Figures SF 4 and SF 8): These figures further illustrate why this behavioral paradigm cannot be considered to only represent “boredom” behaviors. Differences in information seeking, exploratory drives, or decreased perceived aversiveness of the self-administered air puffs may account for some of these behavioral effects. A decrease in movement is very likely to cause a decrease in exploration of every part of the arena. In complete learning coupled to this change in exploration could also cause such exploration to modify the rate of

behaviors that promote repeated airpuff self-stimulation, at least in part.

As this reviewer mentioned, locomotion decreased in the Mus+Bac and naltrexone groups and might have reduced the opportunity to explore the air-puff chamber, at least on the day when these drugs were administered. In this regard, the mice were allowed sufficient time to explore the same chamber before starting the experiment and were fully trained on the relationship between nose pokes and air puffs. Although we cannot strongly argue that the decrease in locomotion had no effect on the results observed in the chamber, the effect of Mus+Bac and naltrexone on the numbers of nose pokes was certainly observed even after the numbers were divided by the distance traveled by the mice. Therefore, we believe that Mus+Bac and naltrexone are considered to reduce nose pokes in the empty chambers.

3-2-2) - SF 4 – This figure, titled “GABA receptor blockade of the AIC does not alter locomotor activity,” is misleading. Muc + bac in AIC does indeed affect locomotion variables presented in panels A and C. Further, these effects may be stronger in empty environments versus enriched ones, but the authors do not fully discuss these nuances. It is also unclear as to why muc +bac in AIC changes the distance traveled in the empty/enriched task but not in the OFT results presented in panel E. - SF 8 – Same issue as SF4 – naltrexone does have effects on locomotion – which again may be stronger in the empty condition. Again – it is confusing that there are no effects of naltrexone on OFT activity.

We have changed the title of Supplementary Figure 8 to "Infusion of GABA receptor agonists into the AIC does not alter locomotor activity in the novel open field test", which is the main point we would like to state here; that is, Mus+Bac does not always reduce motivation in all situations. Apart from this point, the concerns raised by this reviewer are still valid. We also do not exactly know if the effects of Mus+Bac differed between the in-chamber and open-field tests, but we speculate that the open field was a novel environment for the mice, whereas the need for exploration was no longer present in the familiar environment (air puff chamber), which might be why locomotion was reduced by Mus+Bac and naltrexone in the empty chamber.

3-3-1) Fig 1 + 2: Fig 1 and 2 could be combined to more clearly present the behavioral paradigm to readers.

We appreciate this constructive comment. Figures 1 and 2 are strongly interrelated. Therefore, according to this comment, we have decided to merge the two diagrams.

3-3-2) The experimental design includes 4 blocks, but there is no control for behavioral habituation/acclimation to the potential neophobia associated with the airpuff. The four sequential conditions (enriched-1, empty-1, enriched-2, empty-2) are not counterbalanced, and are grouped into two bars (all enriched, or all empty-cage). Might the ordering drive part of the observed effect? This could be examined by showing the mean for all 4 blocks in the bar graphs in Fig1C,D,E and Fig 2C. As it will allow estimation and ruling out of whether e.g. enriched-2 also shows more self-stimulation than enriched-1 due to a gradual upward trend. Most convincing would be if empty-1 showing more self-stimulation than both enriched-1 and enriched-2 when considered separately. If the above requested analyses call the results into question, a more compelling experiment would be to demonstrate that the behavioral effects remain the same if the exposure to the box conditions is reversed (e.g. – empty – enriched – empty – enriched)

This comment points to a shortcoming in our experimental design. It is true that the empty and enriched conditions were not counterbalanced in this study, and it was not an elegant experimental design. Thus, we have repeated the same experiment with a new series (empty-1, enriched-1, empty-2, and enriched-2) with the order swapped. The data are shown in Supplementary Figure 2, confirming again that nose pokes increased under the empty conditions. Additionally, we plotted the mean values for all 4 blocks in the bar graphs in the same figure.

- There is significant discrepancy in the sample sizes between conditions in these two figures, that should be addressed and / or justified via a power analysis of expected effect sizes (e.g., Fig 1, $n = 4$ in “no air puff” vs. $n = 23$ in “air puff” conditions).

This was true. Since a discrepancy in the number of samples may lead to statistically incorrect conclusions, we performed an additional experiment and increased the “no air puff” group to $n = 10$.

- The use of bootstrapping is useful given later claims about non-normal distribution of nose poke behaviors.

The number of nose pokes is not normally distributed, as this reviewer expected; actually, this is why we plotted the data on a logarithmic scale. The results of the normal distribution test are shown in the Methods section (P26, L20), and it is noted that the bootstrap test was performed because of this.

3-4) Fig 3 – nose poking necessitates a stop in locomotion. How much of a drop in HR and increase in HRV would you expect simply from this behavioral change? The ~10 bpm change observed around the nose poke entry appears to be a small effect size. In general, this figure distracts from the rest of the work and may be better suited to be a supplementary figure.

As per this comment, the effect size was small and cannot make a strong statement, so it is not necessary to include the data in the main figures. As mentioned above in the response to a comment of another reviewer, we have eliminated the context of ‘boredom’ from the main argument of this paper. As the data of HR and HRV may still be important, we have decided to move them to Supplementary Figure 6.

3-5) Fig 4 – Panels A,B,F,G seem like useful loss and gain of function experiments, pending interpretational issues related to motivational drives. The manipulation of activity in anterior insular cortex appears to be somewhat similar to manipulations of anterior ‘sweet taste cortex’ by Charles Zuker’s group.

o Jin, H., Fishman, Z.H., Ye, M., Wang, L., and Zuker, C.S. (2021). Top-Down Control of Sweet and Bitter Taste in the Mammalian Brain. *Cell* 184, 257–271.e16. <https://doi.org/10.1016/j.cell.2020.12.014>.

o Wang, L., Gillis-Smith, S., Peng, Y., Zhang, J., Chen, X., Salzman, C.D., Ryba, N.J.P., and Zuker, C.S. (2018). The coding of valence and identity in the mammalian taste system. *Nature* 558, 127–131. <https://doi.org/10.1038/s41586-018-0165-4>

For example, stimulation of this region by the Zuker group has promoted seeking behaviors for aversive stimuli, which seems highly relevant here, and may also modify the interpretation for the role of AIC as a general driver of seeking behaviors in more complex contexts with competing behavioral choices.

Thank you for this valuable comment. As Zuker et al. described, there is a region in the AIC that corresponds to sweet taste, and the activation of this region leads to licking in response to an aversive stimulus (bitter taste), so it is possible that the activation of the AIC promoted the seeking behavior for air puffs in this study as well. This might be mirrored in the behavior of seeking puffs in an empty environment. We have added this discussion in the Discussion section (P9, L34).

3-6-1) The title of this figure appears totally different from the above loss/gain or function: “Gamma oscillations in the AIC increase before nose-poking behavior,” and should be modified. More generally, the discussion focusing on gamma oscillations (panels C-E) in general appears somewhat arbitrary and distracts from the main messages of this paper. In this reviewer’s opinion, it might be best to remove these panels from the paper entirely, or at the least to move to the supplement. Interpretation of such changes in rhythmicity is challenging as it’s likely to occur in many other brain regions as well, and to be related at least in large part to arousal and motor behaviors rather than strictly to learning or behavioral choice.

As this reviewer mentioned, changes in gamma oscillations could certainly occur in other brain regions, and there is no certainty that the changes were specifically related to boredom. Given the shift in the main focus of this paper, we do not claim ‘boredom’ as a result of the gamma oscillation changes. However, because of the link between nose-poke behaviors and gamma oscillations and because of the use of gamma rhythms as a condition for optogenetic stimulation, we believe that the data are still important and need to be retained in the body text to avoid confusing readers. Please note that we performed similar analyses for the other oscillation bands (delta, theta, beta, and alpha), but only the gamma oscillations showed such a specific change. Therefore, we believe that the observed change in gamma oscillations in the AIC has functional significance.

3-6-2) In Fig 4D, please include additional spike rasters (as in SF6C)

This may be a good idea, but unfortunately, adding a raster plot to Fig. 2d is not very legible because the raster is filled with dots due to the different time scales.

3-6-3) Are all opto AIC manipulations done in “empty box” conditions?

We conducted optogenetic AIC manipulation under both empty and enriched conditions. Unlike in the empty chamber, the same optogenetic stimulation did not increase nose pokes in the enriched chamber. The data were not shown in the previous manuscript, but they are now presented in Supplementary Figure 10f. We speculate that even if so-called “boredom” is artificially augmented in the enriched chamber, it is mitigated by playing with toys, so there was no need for the mice to perform nose pokes.

3-7) Fig 5 – As with Figure 4C-E, it is unclear what Figure 5 really adds to the work. Instead, it seems to distract from the main messages. Overall the yield in neurons (N=93) is quite low and the insights gained appear minimal. The decoding accuracy is not that impressive, and it is unclear if neural recordings in AIC are special in any way or if similar results would be seen in most brain areas. Perhaps this figure could be removed or these data could be moved to the supplement prior to Figure 4, to motivate somewhat the use of silencing of insular cortex?

- To claim that the AIC is uniquely involved in this behavior, it would be more compelling to compare AIC activity across nose pokes that are paired with an air puff

versus nose pokes that are not paired with an air puff, and to also record in other brain areas, ideally at the same time.

Thank you for this important comment. Certainly, not much can be said from the spike data, but we think it is still worthwhile to assert at least that neurons in the AIC (leaving out other regions) exhibit nose-poke related firing. The recordings from 93 cells are certainly very small in terms of the amount of data, but conversely, we could say that the fact that predictions can be made from such a small amount of data may suggest the involvement of the AIC.

3-8) Fig 6 –

- The authors claim in the subheading for the manuscript that “Aversive stimuli act as a reward when mice were bored.” Here, they claim that “anticipation of the air puff becomes a reward”. Neither interpretation is entirely compelling. The discussion around salience vs. valence encoding of dopamine release in the VLS is confusing and not entirely consistent with existing literature – e.g. - de Jong et al. 2018 (cited in line 203). As the authors mention, some of the dopamine release may reflect salience rather than reward or reinforcement (see above).

o Also, it is not well substantiated that dopamine release in the VLS reflects “how mice perceive” events (line 198).

Yes, it is true that dopaminergic neurons respond to both reward and aversion. This was why we employed control experiments using pellets and passive air puffs. The results showed an increase in signal in response to pellets and a decrease in signal in response to passive air puffs, suggesting that in our experimental data (at least obtained from the sites in the ventral striatum that we targeted), positive/negative dopamine signals were consistent with reward/aversion. Nevertheless, this comment that we must be cautious in our writing in this regard is very important. We have tried to keep the description factual, for example, by removing the word “reward” from the title.

3-9) Fig 7 – There are concerns with using the term “addiction”.

- This is not a conventional use of the term. Also, the authors appear to define the ‘ramping up’ of self-stimulation compared to a very short initial period of measurement. It’s unclear if this is just the animal getting over neophobia of some sort. For example, would this development of “addiction” behavior still occur in the empty context if the mice were thoroughly habituated to the airpuff self-stimulation in the enriched context? It is not convincing that the dopamine transference to the pre-air-puff time in the “post addiction” state is a feature of addiction versus simply habituation and learning from repeated presentations. Should show comparably timed dopamine recordings from “non-addicted mice”

We thank this reviewer for the very valuable comments and insights. As the reviewer stated, boredom is a more chronic symptom and not one that can be measured on our experimental time scale. If the increase in nose-poke frequency was observed over a longer period of time, the effect might be transitory and disappear thereafter. However, we consider it unlikely that nose pokes increased because the mice overcame neophobia. This is because the number of nose pokes in the addiction state exceeded the number of nose pokes in the blank hole condition, as indicated by the ratio in Figure 1f. These data indicate that when in a state of addiction, mice preferred to take air puffs.

In any case, there is no evidence that the phenomenon we observed here is consistent with addiction in clinical practice. Therefore, in this revised manuscript, we have removed the term ‘addiction’ and have simply referred to the frequent state of nose-poking as an “intense state.”

3-9-3) Pre-treating mice with naltrexone before an exposure to the behavioral paradigm is not similar to how this drug is used in humans, where it is used to treat substance use disorders once they have developed. The clinical relevance is thus not clear. Further, the effects are also consistent with a decreased exploratory drive. Have the authors then tested the same mice without naltrexone to see if the addictive behaviors develop? A more conventional application of naltrexone might be to examine the effects of naltrexone on the behavior and / or ventral striatum dopamine dynamics in mice that are already displaying high levels of air puff self-administration.

Naltrexone is already in clinical use for the treatment of addiction, but we have also decided not to use the term 'addiction' in this manuscript. Interestingly, our data that were not shown in the previous manuscript showed that naltrexone suppressed the dopamine increase immediately before nose pokes. This fact indeed motivated us to hypothesize that naltrexone might inhibit the transition to the intense state. We have included the dopamine data in this revised manuscript (Supplementary Fig. 14).

3-10) It would be useful to put this study in the context of literature in which neuroscientists have described prioritized salience maps of the environment and how these guide exploration (e.g., in parietal cortex). Further, it would be useful to compare the current work to studies of animal preferences for medium-preference fruits in primates depending on whether higher or lower preference fruits are available, corrects in dopamine neuron activity. Might these studies have parallels to the current findings?

It is an interesting and important discussion. The present study suggests that activation of the insular cortex may cause dopamine elevation. Such dopamine elevation may contribute to the formation of what is referred to as a "prioritized salience map" in the parietal cortex (Fectau et al., 2006; Gottlieb et al. Fectau et al., 2006; Gottlieb et al., 2007). In other words, dopamine modification may induce a preference for high-salience locations. This has been added to the Discussion with relevant references cited (P10, L29).

3-11) Fig 6 –

- Incorrect use of the term "imaged" when referring to photometry recordings
- Are the reported light power values of 10 uW for the 465 and 405 LED correct? This seems low and is perhaps may be due to the authors not accounting for the LED duty cycle? Does this represent peak power of average power?
- B: food pellet positive control – please clarify if mice were food restricted
 - o Please add error bars for trace in bottom left
- C,D,E: are these all recorded in the empty cage condition?
 - o D: Please add error bars for traces?

The LED power was indeed 10 uW. We have developed an experimental system using GasAs, which is a much more sensitive photon detector than the widely used PMT. The low laser intensity enabled us to stably monitor dopamine signals for a long time without photobleaching and photodamage. Please refer the detail to our bioRxiv paper titled "The dopamine dip amplitude is a quantitative measure of disappointment (Shikano et al.)" (<https://www.biorxiv.org/content/10.1101/2021.07.23.453499v3>).

3-12) SF 3 – The histology summary for Fig 4 manipulations is nice. However, comparing sagittal and coronal placement for different manipulations is difficult for the

reader.

Thank you for your kind comments. Originally, the sections should be aligned with certain directions to facilitate understanding, but due to the different sizes of electrodes and cannulae, the angles of insertion into the brains were different. Therefore, to confirm these positions, we had to make sections in different directions. Please forgive me for leaving this diagram unchanged as it is. I apologize for the potential confusion.

3-13) SF 7 – The claim that the population distribution of nose poke behavior is bimodal is not compelling based only on the histogram in panel A – this be more compelling to demonstrate this via a statistical test for bimodality. It is unclear that this point is worth pushing too strongly.

We have shown that the bimodal fitting was statistically significant (Supplementary Fig. 12a, bimodality coefficient = 0.75). However, this comment is correct. We merely wanted to show that the distribution has a local minimum at approximately 2.5/min and can be roughly divided into individuals with high and low poke frequencies; it is not necessary to show that the distribution conforms to a strictly bimodal distribution in the statistical sense.

3-1) 5) Please add error bars in Figure 6E and 7E

We have added error bars.

3-14) Please add Deng et al., 2021 Cell citation for insular cortex stimulation VLS dopamine discussion, line 291

Thank you for this suggestion. Because Deng's work addressed reward-related behavior but not place preference, we have added "optogenetic stimulation of a subclass of AIC neurons increases dopamine release in the ventral striatum and promotes motivational vigor even in trials without reward" (P10, L27).

Thank you for your reconsideration of our manuscript.

Sincerely,

Yuji Ikegaya, PhD
Laboratory of Chemical Pharmacology
Graduate School of Pharmaceutical Sciences
The University of Tokyo
7-3-1 Hongo, Bunkyo-ku, Tokyo 113-0033, Japan
Phone: +81-3-5841-4780, Fax: +81-3-5841-4786
E-mail: yuji@ikegaya.jp

REVIEWERS' COMMENTS

Reviewer #1 (Remarks to the Author):

I suspect that I was asked to review this paper given my expertise in boredom. While I fully understand (and agree with) the change in focus here (removing all mention of boredom) I can't pretend I'm not disappointed. In some sense this experimental design had an obvious antecedent in Timothy Wilson's work with humans. So I might advocate for some brief hint at that parallel in the Discussion but this is not absolutely necessary.

The work is fascinating and the authors have given careful consideration to all concerns. So I am happy for it to proceed to publication.

Reviewer #2 (Remarks to the Author):

The authors responded to all comments and criticism of the paper made by the reviewers. However, in doing so, they eliminated the critical aspect of the article, like addiction and boredom. The paper's title did not accurately reflect the findings since they demonstrated that the insular cortex is highly involved in aversive stimulus seeking. Instead, the title suggests mesolimbic dopamine activity is mainly engaged in aversive stimuli seeking without showing cortical DA involvement.

Reviewer #3 (Remarks to the Author):

The revised manuscript is significantly improved, both in the addition of new experiments and, in many places, in the toning down of claims. While many of my concerns are addressed, some still require attention and I also have comments on the new focus on dopamine responses preceding nose poke.

Major comments:

1) The authors now emphasize dopamine (DA) release happening before the nose poke. I'm not sure this is a title or emphasis that best captures the overall manuscript, and I think it requires further context

relative to existing literature: It's just as true that it likely happens during the decision to approach the nose poke. The notion that DA release occurs during behavior is quite common, even in the absence of overt rewards. Certainly this is true in dorsal striatum perhaps less true in some parts of NAc (e.g. Dombek <https://pubmed.ncbi.nlm.nih.gov/27398617/>; <https://www.nature.com/articles/s41586-022-05611-2>; these should be discussed). In general, the authors should analyze or at least discuss other behaviors that precede each actual poke, to understand if dopamine actually precedes the start of the behavioral sequence leading to nose poke, or not.

2) The response to 3-1, that the mice have experienced airpuffs and thus "an air puff was not an 'unexpected' stimulus for mice", may not be correct. It is unclear if the mice have learned the precise relationship between their degree of head entry into the nosepoke vs the moment air puff. Moreover, this expectation of timing may change gradually over sessions.

- Also, it would be worth speculating whether the nose poking is simply part of an action program searching for escape from the cage (regardless of aversive airpuffs)? If so, the increased dopamine may reflect reinforcement of actions that serve to promote searching for an escape from the cage.

-

3) Regarding 3-7, the quality of insula electrophysiology data: I disagree with the authors here. The data are quite noisy regardless of any statistical tests, and cause more harm than good to the overall trustworthiness and interpretability of the manuscript. I strongly suggest removing it or, at the very least, moving it to the supplement. If the goal is to prove nose-poke related firing, one can focus on individual neurons and do circular shifting and show that individual neurons modulate their firing surrounding nose-pokes. If this is too noisy to see on individual pokes, that's a problem. Few readers will be impressed that there is such a weak and noisy in activity change visible only at the population level.

Minor comments:

3-11: The authors still use 'imaged' when there is no image, in legend for Figure 4.

3-13: regarding bimodal fitting: this still seems unlikely. If this is a real bimodal distribution, it should show up on each subset when splitting the data into two equal subsets of values. A bimodality coefficient of .75 doesn't help make this statement convincing. It's fine if the authors use an arbitrary threshold, but better to admit it's arbitrary.

Comments and Answers

Reviewer #3

Thank you for the positive evaluations, which have encouraged us to resubmit this manuscript. We have revised our manuscript in accordance with your comments. Our point-by-point responses are as follows:

1) The authors now emphasize dopamine (DA) release happening before the nose poke. I'm not sure this is a title or emphasis that best captures the overall manuscript, and I think it requires further context relative to existing literature: It's just as true that it likely happens during the decision to approach the nose poke. The notion that DA release occurs during behavior is quite common, even in the absence of overt rewards. Certainly this is true in dorsal striatum perhaps less true in some parts of NAc (e.g. Dombeck <https://pubmed.ncbi.nlm.nih.gov/27398617/>; <https://www.nature.com/articles/s41586-022-05611-2>; these should be discussed). In general, the authors should analyze or at least discuss other behaviors that precede each actual poke, to understand if dopamine actually precedes the start of the behavioral sequence leading to nose poke, or not.

Thank you for your valuable suggestions. We strongly agree that the possibility of a rise in dopamine levels occurring when determining whether or not to approach the poke hole is a plausible idea, based on past literature. However, we did not observe a dopamine increase before the nose pokes under the w/o air-puff condition (Fig. 3d, e), which suggests that the dopamine increase is not simply a result of a general increase in motor activity or decision-making. Additionally, we calculated the movement velocities of the mice as a measure of pre-poke behavior and examined the relationship between pre-poke velocity and dopamine levels. We found no significant correlation between the two, indicating that the dopamine increase was unlikely to be preceded by the mice's movement or preparation for poke behaviors. We have added this point to the Results section (P8, L28) and included a new Supplementary Figure 13. We appreciate this comment as it further clarifies our findings.

2-1) The response to 3-1, that the mice have experienced airpuffs and thus "an air puff was not an 'unexpected' stimulus for mice", may not be correct. It is unclear if the mice have learned the precise relationship between their degree of head entry into the nosepoke vs the moment air puff. Moreover, this expectation of timing may change gradually over sessions.

Thank you for your helpful comment. We took your suggestion seriously and attempted to investigate how dopamine levels change over sessions by dividing the nose poke series into four equal segments for each mouse and comparing them between the first and last quartiles. We expected that the pre-poke dopamine increase would gradually emerge as the mice learned the "relationship between their degree of head entry into the nose poke vs. the moment air puff." However, we did not find a significant change in dopamine levels in either pre-, during- or post-period. This lack of a time-dependent change in dopamine further confirms that the mice had already learned the relationship between poking and the resulting air puffs during the pre-training session.

2-2) Also, it would be worth speculating whether the nose poking is simply part of an action program searching for escape from the cage (regardless of aversive airpuffs)? If so, the increased dopamine may reflect reinforcement of actions that serve to promote searching for an escape from the cage.

Thank you for pointing out a point we missed in our earlier manuscript. Although the hole size (1.5 cm in diameter) is not large enough for mice to escape (therefore escape behavior did not directly contribute to the poke behavior), we cannot completely rule out the possibility that dopamine increased due to the reinforcement of behavior prompting the mice to escape from the cage. However, poke frequencies increased more under the w/o air puff condition than under the w/air puff condition (Fig. 1e), we believe that we cannot attribute all the phenomena we observed solely to escape behavior.

3) Regarding 3-7, the quality of insula electrophysiology data: I disagree with the authors here. The data are quite noisy regardless of any statistical tests, and cause more harm than good to the overall trustworthiness and interpretability of the manuscript. I strongly suggest removing it or, at the very least, moving it to the supplement. If the goal is to prove nose-poke related firing, one can focus on individual neurons and do circular shifting and show that individual neurons modulate their firing surrounding nose-pokes. If this is too noisy to see on individual pokes, that's a problem. Few readers will be impressed that there is such a weak and noisy in activity change visible only at the population level.

Thank you for providing valuable feedback on our manuscript. We appreciate your concerns regarding the noise level in our data. We have decided to move this figure to the supplementary section and make the required changes to the manuscript. Additionally, we appreciate your suggestion to focus on individual neurons and use circular shifting to demonstrate nose-poke-related firing. We appreciate your valuable input, which will help us to determine the direction of our future research.

4) 3-11: The authors still use 'imaged' when there is no image, in legend for Figure 4.

Thank you for your thoughtful comments. We agree that this term is not the best choice of wording and have revised it to "optically recorded" (P31, L4).

5) 3-13: regarding bimodal fitting: this still seems unlikely. If this is a real bimodal distribution, it should show up on each subset when splitting the data into two equal subsets of values. A bimodality coefficient of .75 doesn't help make this statement convincing. It's fine if the authors use an arbitrary threshold, but better to admit it's arbitrary.

Thank you for your insightful comment. As you pointed out, we cannot say a high bimodality coefficient does not guarantee the bimodality of the distribution. Therefore, we have revised the manuscript to state that we choose the local minimum as the threshold of the intense state. This definition is more straightforward.